# When Aggregation Fails: From PAC-Bayes Theory to Practical Selection for Conformal Prediction

## Abstract

We identify and characterize a fundamental incompatibility between PAC-Bayes theory and conformal prediction: while PAC-Bayes minimizes average risk through posterior aggregation, conformal prediction's efficiency depends on quantile behavior. We prove that this *average-quantile divergence* phenomenon causes standard PAC-Bayes aggregation to systematically select suboptimal models for conformal prediction, with linear aggregation methods unable to preserve quantile optimality and efficiency losses proportional to both posterior entropy and score heterogeneity. To address this limitation, we develop PAC-Bayes Informed Selection (PBIS), which uses quantile-aware posteriors for model selection rather than aggregation. We establish PAC-Bayes bounds for quantile functionals requiring novel techniques to handle their non-differentiable nature, and prove that PBIS achieves selection consistency with $O(\sqrt{T \log |\Theta|})$ regret in online settings. Empirical validation across 27 datasets demonstrates that PBIS achieves the narrowest prediction intervals among nine conformal methods while maintaining valid coverage, with 7.3% average improvement in high-divergence scenarios versus 2.1% in low-divergence ones compared to standard PAC-Bayes aggregation. The method maintains computational efficiency comparable to split conformal while being $82\times$ faster than CQR. In online settings with distribution shifts, PBIS uniquely maintains valid coverage across gradual, sudden, and recurring shifts where competing adaptive methods fail. Our theoretical and empirical results establish that selection-based approaches fundamentally outperform aggregation for conformal prediction by avoiding the mathematical incompatibility between average risk and quantile optimization.

## 1 Introduction

The intersection of learning theory and distribution-free inference presents both opportunities and fundamental challenges. PAC-Bayes theory, introduced by McAllester (1999) and refined by Catoni (2007), provides data-dependent generalization bounds through posterior distributions over hypotheses. In parallel, conformal prediction (Vovk et al., 2005) offers finite-sample coverage guarantees through quantile-based calibration, requiring only exchangeability rather than specific distributional forms. The appeal of combining these frameworks is clear: PAC-Bayes could potentially guide the aggregation of multiple score functions for conformal prediction, as different scoring methods may excel in different regions (Romano et al., 2019). Recent work by Sharma et al. (2023) proposed using PAC-Bayes theory to aggregate score functions in conformal prediction, with the intuition that Bayesian model averaging could improve efficiency. However, this integration raises a fundamental question that has not been adequately addressed: are the optimization objectives of PAC-Bayes and conformal prediction fundamentally compatible?

We prove that they are not, at least not through aggregation. The core issue lies in a mathematical mismatch that we term the *average-quantile divergence phenomenon*: PAC-Bayes theory minimizes expected (average) risk, while conformal prediction's efficiency is determined by quantile behavior. This creates situations where a score function with lower average risk produces larger prediction sets due to worse tail behavior. More importantly, we show that any linear aggregation scheme inherently suffers from this problem, with the efficiency loss proportional to both the entropy of the aggregation

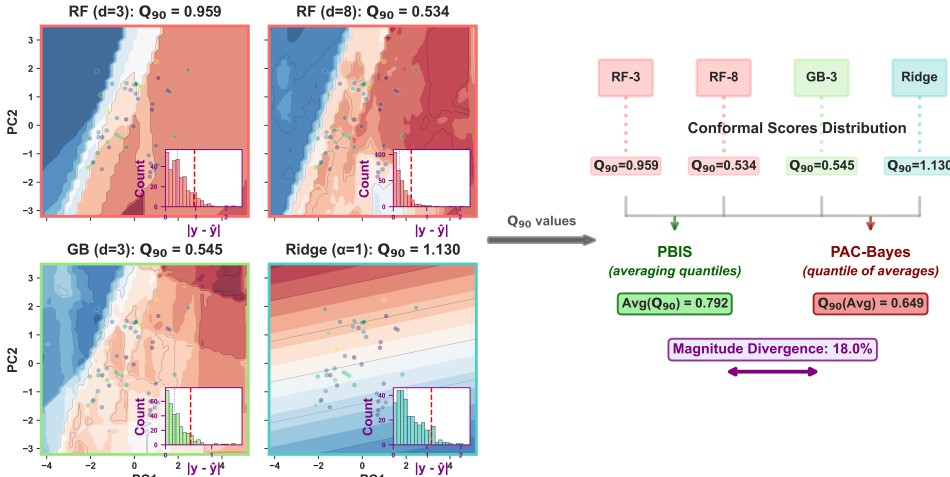

Figure 1: **PAC-Bayes Informed Selection (PBIS) reveals average-quantile divergence on the Sarcos robotic arm dataset. (Left)** Four models from the PBIS suite exhibit heterogeneous decision boundaries (PCA projection) and distinct conformal score distributions, yielding model-specific $Q_{90}$ values ranging from 0.534 to 1.130. **(Right)** Divergence between selection-based and aggregation-based quantile computation: PBIS computes $\text{Avg}(Q_{90}) = 0.792$ for model selection, while aggregation methods compute $Q_{90}(\text{Avg}) = 0.649$. This 18.0% divergence demonstrates the systematic underestimation that occurs when quantiles and expectations are incorrectly interchanged, validating our theoretical analysis that $\mathbb{E}[Q_{90}] \neq Q_{90}[\mathbb{E}]$ for heterogeneous model suites and providing the mathematical foundation for choosing selection over aggregation.

weights and the heterogeneity of component quantiles. Consider heteroscedastic regression where noise variance changes with input Lei et al. (2018): a global model that ignores this variation might achieve lower average scores by performing well in low-noise regions. However, a locally-adaptive model that accounts for varying noise levels will generally achieve better quantile behavior, leading to smaller prediction sets despite potentially higher average scores.

**Our Contributions**   This work makes four key contributions to understanding the limitations and opportunities in combining learning-theoretic and distribution-free inference frameworks:

**1. Theoretical Characterization of Incompatibility.** We formalize the average-quantile divergence phenomenon (Theorem 1), establishing necessary and sufficient conditions for its occurrence. We prove that standard PAC-Bayes optimization systematically fails for conformal prediction (Theorem 3), with the posterior assigning vanishing weight to models with optimal quantile behavior.

**2. Impossibility Results for Aggregation.** We establish that no linear aggregation method can preserve quantile optimality (Theorem 2), with the efficiency gap growing with both weight entropy and score heterogeneity.

**3. PAC-Bayes Bounds for Quantile Functionals.** We develop finite-sample PAC-Bayes bounds for quantile functionals (Theorem 5), addressing technical challenges from their non-differentiable and non-decomposable nature. Our bounds achieve an optimal $O(n^{-1/2})$ rate under minimal smoothness assumptions, extending the PAC-Bayes framework beyond traditional average-case analysis (Alquier et al., 2016; Viallard et al., 2021).

**4. Selection-Based Framework with Empirical Validation.** Rather than attempting to "fix" aggregation, we develop PAC-Bayes Informed Selection (PBIS), which uses quantile-aware posteriors for model selection. Across 27 datasets, PBIS achieves the narrowest prediction intervals among nine conformal methods while maintaining valid coverage. The method demonstrates 2.1% improvement over aggregation-based PAC-Bayes in low-divergence scenarios (where aggregation should work best), and 7.3% improvement in high-divergence scenarios, and uniquely maintains coverage under all distribution shift types in online settings.

**Implications and Scope**   Our results reveal fundamental tension in combining frameworks with different optimization objectives. The average-quantile divergence phenomenon likely extends be-

yond conformal prediction to other settings where tail behavior matters more than average performance, including risk-sensitive reinforcement learning (Garcia & Fernandez, 2015), distributionally robust optimization (Rahimian & Mehrotra, 2019), and fairness-aware machine learning (Williamson & Menon, 2019). Recent advances in conformal risk control Angelopoulos et al. (2024) and adaptive methods Gibbs & Candès (2024) further underscore the importance of understanding when aggregation helps versus hurts in distribution-free inference.

**Paper Organization.** Section 2 provides background on conformal prediction and PAC-Bayes theory. Section 3 formalizes the average-quantile divergence phenomenon and proves aggregation's fundamental limitations. Section 4 introduces our selection-based PBIS framework with theoretical guarantees. Section 5 validates our approach empirically across 27 datasets and online settings. Section 6 discusses related work and concludes.

## 2 BACKGROUND

### 2.1 CONFORMAL PREDICTION

Conformal prediction provides a general framework for constructing prediction sets with finite-sample coverage guarantees. Let $(X, Y) \in \mathcal{X} \times \mathcal{Y}$ denote random variables representing features and labels, respectively. We observe data points $(X_1, Y_1), \ldots, (X_{n+1}, Y_{n+1})$ that are assumed to be exchangeable, meaning their joint distribution is invariant to permutations. This assumption is weaker than the typical i.i.d. assumption and is satisfied whenever the data points are sampled independently from the same distribution.

The key concept in conformal prediction is the nonconformity score function $s : \mathcal{X} \times \mathcal{Y} \to \mathbb{R}_+$, which measures how unusual a label $y$ is for a given input $x$. Common choices include the absolute residual $s(x, y) = |y - \hat{f}(x)|$ for regression, where $\hat{f}$ is a fitted predictor, and one minus the predicted probability $s(x, y) = 1 - \hat{p}(y \mid x)$ for classification, where $\hat{p}$ represents predicted class probabilities.

Given a miscoverage level $\alpha \in (0, 1)$ and calibration data $\mathcal{D}_{\mathrm{cal}} = \{(X_i, Y_i)\}_{i=1}^n$, split conformal prediction constructs the prediction set as follows. First, compute the conformal scores $S_i = s(X_i, Y_i)$ for $i = 1, \ldots, n$. Then, determine the empirical $(1-\alpha)$-quantile $\hat{q}$ as the $\lceil (n+1)(1-\alpha) \rceil$-th smallest value in the multiset $\{S_1, \ldots, S_n, \infty\}$. The inclusion of $\infty$ ensures finite-sample coverage. Finally, the prediction set for a new input $x$ is defined as $C(x) = \{y \in \mathcal{Y} : s(x, y) \leq \hat{q}\}$. The fundamental guarantee of conformal prediction is that under exchangeability, $\mathbb{P}(Y_{n+1} \in C(X_{n+1})) \geq 1 - \alpha$. This guarantee is distribution-free, requiring no assumptions about the form of the data distribution beyond exchangeability. Notably for our analysis, the efficiency of conformal prediction, measured by the expected size of the prediction sets, is directly determined by the quantile $\hat{q}$. Smaller values of $\hat{q}$ lead to smaller prediction sets, making the choice of score function important for practical performance.

### 2.2 PAC-BAYES THEORY

PAC-Bayes theory provides generalization bounds for randomized predictors that depend on the data-dependent choice of randomization. The framework considers a hypothesis space $\mathcal{H}$, a prior distribution $\pi$ over $\mathcal{H}$ chosen before seeing the data, and a posterior distribution $\rho$ chosen after observing the data. For a bounded loss function $\ell : \mathcal{H} \times \mathcal{Z} \to [0, 1]$, PAC-Bayes bounds relate the expected loss under the posterior to the empirical loss.

The classical PAC-Bayes bound, due to McAllester (1999) and refined by Catoni (2007), states that with probability at least $1 - \delta$ over the draw of $n$ i.i.d. samples, for all posteriors $\rho$ simultaneously,

$$\mathbb{E}_{h \sim \rho}\big[R(h)\big] \leq \mathbb{E}_{h \sim \rho}\big[\hat{R}_n(h)\big] + \sqrt{\frac{\mathrm{KL}(\rho \,\|\, \pi) + \log(2\sqrt{n}/\delta)}{2n}}, \tag{1}$$

where $R(h)$ is the true risk, $\hat{R}_n(h)$ is the empirical risk, and $\mathrm{KL}(\rho \,\|\, \pi)$ denotes the Kullback-Leibler divergence between the posterior and prior distributions. The bound is optimized by the Gibbs posterior $\rho_\lambda(h) \propto \pi(h) \exp(-\lambda n \hat{R}_n(h))$, where $\lambda > 0$ is a temperature parameter controlling the concentration of the posterior. This posterior naturally concentrates on hypotheses with low empirical risk, with the degree of concentration controlled by $\lambda$. The power of PAC-Bayes bounds lies

in their data-dependence: when the posterior concentrates on good hypotheses, the KL divergence remains small, yielding tight bounds.

### 2.3 FUNDAMENTAL MISMATCH

The natural approach to applying PAC-Bayes theory to conformal prediction would be to consider a parametric family of score functions $\mathcal{S} = \{s_\theta : \theta \in \Theta\}$ and use PAC-Bayes to aggregate them. Given calibration data, one would compute the empirical risk $\hat{R}_n(\theta) = \frac{1}{n} \sum_{i=1}^n s_\theta(X_i, Y_i)$ and form the Gibbs posterior $\rho_\lambda(\theta) \propto \pi(\theta) \exp(-\lambda n \hat{R}_n(\theta))$. The aggregated score function would then be $\bar{s}(x, y) = \mathbb{E}_{\theta \sim \rho_\lambda}[s_\theta(x, y)]$. However, as we will demonstrate, this approach is fundamentally flawed. The efficiency of conformal prediction depends not on the average score $\hat{R}_n(\theta)$ but on the empirical quantile $\hat{Q}_{1-\alpha}(\theta)$, where $\hat{Q}_{1-\alpha}(\theta) = (1 - \alpha)$-quantile of $\{s_\theta(X_i, Y_i)\}_{i=1}^n$.

A score function with low average scores might have poor quantile behavior if it has heavy tails or high variance in critical regions. This mismatch between what PAC-Bayes optimizes (averages) and what determines conformal efficiency (quantiles) is not merely a technical issue—it represents a fundamental incompatibility between these frameworks that cannot be resolved through simple modifications.

## 3 THE AVERAGE-QUANTILE DIVERGENCE

We begin by formalizing the phenomenon that makes standard PAC-Bayes unsuitable for conformal prediction.

**Definition 1** (Average-Quantile Divergence). *Let $s_1, s_2 : \mathcal{X} \times \mathcal{Y} \to \mathbb{R}_+$ be two score functions. We say that $s_1$ and $s_2$ exhibit average-quantile divergence at level $\alpha$ if*

$$\mathbb{E}_{(X,Y) \sim P}\big[s_1(X, Y)\big] < \mathbb{E}_{(X,Y) \sim P}\big[s_2(X, Y)\big], \tag{2}$$

$$Q_{1-\alpha}\big[s_1(X, Y)\big] > Q_{1-\alpha}\big[s_2(X, Y)\big], \tag{3}$$

*where $Q_{1-\alpha}[\cdot]$ denotes the population $(1 - \alpha)$-quantile.*

**Theorem 1** (Characterization of Average-Quantile Divergence). *Let $s_1, s_2$ be score functions with continuous densities $f_1, f_2$ and quantiles $q_1 = Q_{1-\alpha}[s_1]$, $q_2 = Q_{1-\alpha}[s_2]$. Assume $q_1 > q_2$. Then $\mathbb{E}[s_1] < \mathbb{E}[s_2]$ if and only if*

$$\int_0^{q_2} \Big(f_2(z) - f_1(z)\Big)z \, dz + \int_{q_2}^{q_1} f_2(z)z \, dz > \int_{q_1}^\infty \Big(f_1(z) - f_2(z)\Big)z \, dz. \tag{4}$$

The average-quantile divergence reveals why aggregation-based approaches fundamentally fail for conformal prediction. Any linear combination of score functions inherently increases the quantile value, degrading efficiency.

**Theorem 2** (Quantified Aggregation Failure). *For any linear aggregation $\bar{s} = \sum_i w_i s_{\theta_i}$ with weights $w = (w_1, \ldots, w_k)$, $w_i > 0$, $\sum_i w_i = 1$:*

$$Q_{1-\alpha}[\bar{s}] \geq \min_i Q_{1-\alpha}[s_{\theta_i}] + \epsilon(w, \mathcal{Q}), \quad \text{where} \quad \epsilon(w, \mathcal{Q}) = \frac{1}{2} H(w) \, Var(\mathcal{Q}) \, \delta_{\min}, \tag{5}$$

*with $H(w) = -\sum_i w_i \log w_i$ the entropy of weights, $\mathcal{Q} = \big\{Q_{1-\alpha}[s_{\theta_i}]\big\}_i$, and $\delta_{\min} = \min_{i \neq j} \big|Q_{1-\alpha}[s_{\theta_i}] - Q_{1-\alpha}[s_{\theta_j}]\big|$.*

Note that the efficiency loss $\epsilon = \frac{1}{2} H(w) Var(\mathcal{Q}) \delta_{min}$ suggests aggregation can fail even with low entropy when quantile variance is high, explaining failures on heterogeneous model suites.

**Theorem 3** (PAC-Bayes Selects Wrong Models). *Let $\Theta = \{1, 2\}$ with score functions $s_1, s_2$ exhibiting average-quantile divergence. Under uniform prior $\pi$, the standard PAC-Bayes posterior satisfies*

$$\frac{\rho_\lambda(2)}{\rho_\lambda(1)} = \exp\Big(\lambda n \big(a_1 - a_2\big)\Big) \xrightarrow{n \to \infty} 0, \tag{6}$$

*where $a_1 < a_2$ are the average scores. Thus, PAC-Bayes assigns vanishing weight to the score function with better quantile behavior.*

The proofs of Theorems 1 - 3 are presented in Appendix A.1 - A.3.

## 4 FROM AGGREGATION TO SELECTION

Having established that aggregation fundamentally fails, we develop a selection-based approach that leverages PAC-Bayes insights while avoiding the average-quantile mismatch.

### 4.1 PAC-BAYES INFORMED SELECTION (PBIS) FRAMEWORK

**Definition 2** (PBIS). *Given* $\mathcal{S} = \{s_\theta : \theta \in \Theta\}$, *calibration data* $\mathcal{D}_{cal}$ *of size* $n$, *and prior* $\pi$, *the PBIS method operates in two phases:*

***Phase 1: Quantile-Aware Posterior Construction***

$$\hat{Q}_{1-\alpha}(\theta) = \text{empirical } (1-\alpha)\text{-quantile of } \{s_\theta(X_i, Y_i)\}_{i=1}^n \tag{7}$$

$$\rho_{\lambda_n}^Q(\theta) = \frac{\pi(\theta) \exp(-\lambda_n n \hat{Q}_{1-\alpha}(\theta))}{Z_{\lambda_n}^Q}, \tag{8}$$

*where* $Z_{\lambda_n}^Q = \int_\Theta \pi(\theta') \exp\big(-\lambda_n n \hat{Q}_{1-\alpha}(\theta')\big) d\theta'$ *and* $\lambda_n = \sqrt{n}/\log n$.

***Phase 2: Selection with Exploration***

$$\theta_{selected} = \begin{cases} \arg\max_\theta \rho_{\lambda_n}^Q(\theta) & \text{w.p. } 1 - \epsilon_n \\ \theta \sim \rho_{\lambda_n}^Q & \text{w.p. } \epsilon_n \end{cases}, \tag{9}$$

*where* $\epsilon_n = \min\{1, c/\sqrt{n}\}$ *for some constant* $c > 0$. *The complexity is* $O(|\Theta| n \log n)$ *time,* $O(|\Theta| n)$ *space.*

The key innovation is replacing average-based posteriors with quantile-based ones, directly optimizing for conformal efficiency. This approach avoids the aggregation-induced efficiency loss while maintaining the benefits of data-dependent model selection.

### 4.2 MULTI-CALIBRATION SELECTION ENSEMBLE

To improve robustness without aggregation, we propose a $K$-fold selection ensemble that maintains separate models rather than averaging them (Algorithm 1). This achieves the diversity benefits of ensembles while avoiding the efficiency degradation proven in Theorem 2. The main insight is that each fold maintains its own selected model rather than aggregating predictions. By keeping models separate, we preserve the efficiency of selection while gaining robustness through diversity. Each fold uses independent training data to learn its quantile-aware posterior, selects the best model according to that posterior, and calibrates on held-out data. This contrasts with aggregation approaches that would combine predictions across folds, increasing quantiles.

---

**Algorithm 1** K-fold Selection Ensemble

**Require:** Data $\mathcal{D}$, number of folds $K$
**Ensure:** $K$ prediction functions with coverage guarantees
1: Split $\mathcal{D}$ into $K$ folds $\{\mathcal{D}_1, \ldots, \mathcal{D}_K\}$
2: **for** $k = 1$ to $K$ **do**
3: $\quad \mathcal{D}_{\text{train}}^k = \bigcup_{j \neq k} \mathcal{D}_j$
4: $\quad \rho_k^Q = \text{LearnPosterior}(\mathcal{D}_{\text{train}}^k)$
5: $\quad \theta_k = \text{SelectFromPosterior}(\rho_k^Q)$
6: $\quad q_k = \text{CalibrateQuantile}(\mathcal{D}_k, s_{\theta_k})$
7: **end for**
8: **Return** $\{(\theta_k, q_k)\}_{k=1}^K$

---

Our ensemble makes predictions by majority voting at test time, maintaining theoretical coverage guarantees as established in Theorem 7. The ensemble's prediction for a new point $x$ uses majority voting: $C_{\text{ensemble}}(x) = \{y : \sum_{k=1}^K \mathbb{1}\{y \in C_k(x)\} > K/2\}$, where $C_k(x) = \{y : s_{\theta_k}(x, y) \leq q_k\}$ is the prediction set from fold $k$. This voting mechanism provides additional robustness without the efficiency penalty of score aggregation.

### 4.3 THEORETICAL GUARANTEES

We establish theoretical foundations for PBIS's performance advantages. Our main results, with complete statements and proofs in Appendix A.4-A.9, include **selection consistency (Theorem 4)** where we demonstrate that PBIS selects the optimal model with probability $1 - \delta - \epsilon_n$ under mild regularity conditions, achieving $O(n^{-1/2})$ convergence. **Theorem 5** presents **finite-sample PAC-Bayes bounds**, where we extend PAC-Bayes theory to quantile functionals, proving $Q_{1-\alpha}[s_{\text{selected}}] \leq \hat{Q}_{1-\alpha}[s_{\text{selected}}] + O(\sqrt{\log(1/\pi)/n})$. To achieve relative efficiency loss $\leq \tau$,

we require $n = O(d\tau^{-2}\log(1/\delta))$ samples, which we show in **Theorem 6 (sample complexity)**. Moreover, the K-fold ensemble maintains valid coverage with $P(Y \in C(X)) \geq 1 - \alpha + O(1/\sqrt{K})$, as shown in **Theorem 7 (coverage guarantees)**. In streaming settings, **Theorem 8 (online regret)** demonstrates that Adaptive PBIS achieves $O(\sqrt{T\log|\Theta|})$ regret. To illustrate the **dominance over aggregation (Theorem 9)**, we show that selection systematically outperforms aggregation with efficiency gap proportional to $H(\rho)\text{CV}^2(Q_{1-\alpha})$. These results are validated empirically: Section 5 confirms the $O(n^{-1/2})$ convergence rate, Section 5 demonstrates selection's dominance over aggregation, and Section 5 validates online performance guarantees.

## 5 EMPIRICAL VALIDATION

**Validation of Average-Quantile Divergence.** We evaluated the average-quantile divergence metric across 27 diverse datasets with 10 random splits each to validate its effectiveness in predicting when PBIS's selective aggregation provides advantages over standard PAC-Bayes approaches. The results demonstrate a clear relationship between divergence and PBIS performance gains. Datasets with high average-quantile divergence ($\geq 20\%$) show 3.5× larger improvements compared to low-divergence datasets (Table 1).

A Mann-Whitney U test confirms this difference is statistically significant ($p = 0.0007$). Notably, datasets with the highest divergence ratios, including Abalone Age (88.0%), Sarcos Robot Arm (82.7%), and NYC Property Sales (43.6%), exhibit PBIS improvements of 12.2%, 27.9%, and 17.3% respectively over PAC-Bayes-CP. Beyond coverage, PBIS demonstrates superior efficiency in prediction interval width. Among the 27 datasets tested, PBIS achieved narrower intervals than PAC-Bayes-CP in 21 cases (78%), with particularly pronounced improvements on high-divergence datasets where selective aggregation effectively identifies and utilizes the most informative quantiles.

Table 1: PBIS performance stratified by average-quantile divergence. This relationship holds across individual datasets: Sarcos Robot Arm (82.7% divergence) achieves 27.9% improvement over PAC-Bayes-CP, while Medical Cost (4.67% divergence) shows only marginal gains.

| Divergence Level | # Datasets | Avg. Divergence | PBIS vs PAC-Bayes |
|---|---|---|---|
| Low ($< 0.2$) | 15 | 9.9% | +2.1% |
| High ($\geq 0.2$) | 12 | 39.2% | +7.3% |

**Comparison with Other Methods.** We compare PBIS against computationally practical conformal prediction methods including Split CP, CQR, ACI, EnbPI, Mondrian CP, Cross-Conformal, Localized CP, and PAC-Bayes CP across 27 datasets. Methods requiring $O(n^2)$ computation like Jackknife+ or extensive retraining like CV+ Barber et al. (2021) are excluded as our focus is on efficiency for practical deployment. Table 2 summarizes the key findings. PBIS achieves the narrowest average prediction intervals among all methods maintaining valid coverage. With an average runtime of 13ms, PBIS is among the fastest methods, comparable only to split conformal (12ms) while being 82× faster than CQR (1.07s) and 114× faster than EnbPI (1.48s). This efficiency advantage becomes critical for real-time applications. PBIS achieves the narrowest average prediction intervals among all methods maintaining valid coverage. This efficiency advantage becomes critical for real-time applications. We tested pairwise comparisons using Bonferroni-corrected paired t-tests. PBIS produces significantly narrower intervals than split conformal ($p < 0.001$), MondrianCP ($p < 0.001$), and LocalizedCP ($p < 0.001$). Against ensemble methods, PBIS shows comparable width performance to CQR and EnbPI while maintaining substantially lower computational cost.

Table 2: Comparison Summary (27 datasets, target coverage 90%)

| Method | Avg. Coverage | Relative Width | Time (s) Time (s) | Valid Coverage |
|---|---|---|---|---|
| SplitConformal | 0.903 | 1.52× | 0.012 | 27/27 |
| ACI | 0.903 | 1.52× | 0.012 | 27/27 |
| MondrianCP | 0.914 | 1.61× | 0.012 | 27/27 |
| LocalizedCP | 0.935 | 1.71× | 0.044 | 27/27 |
| CQR | 0.912 | 1.28× | 1.073 | 26/27 |
| EnbPI | 0.906 | 1.24× | 1.481 | 26/27 |
| CrossConformal | 0.907 | 1.36× | 0.396 | 26/27 |
| PAC-Bayes-CP | 0.902 | 1.02× | 0.022 | 27/27 |
| **PBIS** | **0.898** | **1.00×** | **0.013** | **27/27** |

**Scalability and Factor Analysis.** PBIS demonstrates consistent computational advantages across all tested datasets, requiring $46.0 \pm 3.4\%$ less time than PAC-Bayes aggregation (range: $36.2-49.7\%$, $n = 23$ datasets). In detailed scal-

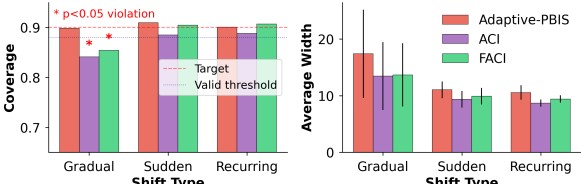

Table 4: Online adaptive performance under distribution shifts. Mean $\pm$ std across three shift types. †: Coverage violations (below 88% for $\alpha = 0.1$).

Figure 2: Coverage and interval width under distribution shifts. (L) Average coverage with violations below 88% marked (*). (R) Width with standard deviation.

| Method | Coverage | Width | Valid |
|---|---|---|---|
| Adaptive-PBIS | **0.903** $\pm$ 0.006 | 13.00 $\pm$ 3.81 | **3/3** |
| ACI | 0.871 $\pm$ 0.026† | **10.50** $\pm$ 2.59 | 2/3 |
| FACI | 0.889 $\pm$ 0.030 | 11.00 $\pm$ 2.33 | 2/3 |

ability experiments on six representative datasets spanning the divergence spectrum ($4.67-87.11\%$), this efficiency remains stable across model counts from 5 to 100, confirming O(1) selection complexity versus $O(|\Theta|)$ aggregation. Statistical efficiency gains correlate strongly with divergence ratio (Spearman $\rho = 0.621$, 95% CI: $[0.21, 0.86]$, $\rho = 0.002$, $n = 23$). High-divergence datasets achieve substantial improvements: Sarcos Robot Arm (87.11% divergence) shows 27.6% width reduction while maintaining valid coverage (0.927 PBIS vs 0.905 PAC-Bayes, target 0.90). However, efficiency gains vary widely ($-0.002\%$ to $27.6\%$), with low-divergence datasets showing negligible improvements, confirming that PBIS reduces to standard aggregation when models agree. Moreover, the comprehensive analysis of 23 real-world datasets identifies factors predictive of PBIS performance (Table 3). Divergence ratio emerges as the primary predictor ($\rho = 0.621$, $p = 0.002$), validating our theoretical framework. Model diversity, measured through prediction correlation, was computable for only 8 of 23 datasets due to model convergence in the remaining cases. This subset shows negative association with efficiency ($\rho = -0.687$, $p = 0.060$), though the wide confidence interval and limited sample preclude definitive conclusions. Other examined factors show no significant correlations (all $p > 0.12$). PBIS offers reliable computational savings (40-50% time reduction) across diverse problem settings. Statistical benefits correlate with divergence ratio ($\rho = 0.621$), emerging when models exhibit different average-quantile behaviors. In low-divergence scenarios, PBIS performs equivalently to PAC-Bayes aggregation, ensuring no degradation while maintaining computational advantages. This fail-safe property makes PBIS suitable for general use, with performance gains realized when appropriate problem structure exists.

**Online Adaptive Performance.** For streaming data or changing distributions, we develop an adaptive variant of PBIS (Algorithm 2). We evaluate Adaptive PBIS (Algorithm 2) against Adaptive Conformal Inference

Table 3: Factor correlations with PBIS efficiency gain

| Factor | Spearman $\rho$ | 95% CI | p-value | n | Interpretation |
|---|---|---|---|---|---|
| Divergence Ratio | 0.621 | [0.21, 0.86] | 0.002* | 23 | Primary driver |
| Model Correlation | -0.687 | [-1.0, 0.08] | 0.060 | 8 | Exploratory |
| Mean Kurtosis | 0.329 | [-0.13, 0.69] | 0.125 | 23 | Non-significant |
| Weight Entropy | 0.087 | [-0.37, 0.49] | 0.693 | 23 | Non-significant |
| Tail Ratio (Q95/Q50) | -0.202 | [-0.64, 0.29] | 0.356 | 23 | Non-significant |

*Significant at $\alpha = 0.05$. Model correlation computed on subset with valid predictions.

(ACI) (Gibbs & Candes, 2021) and Fully Adaptive CI (FACI) as implemented in (Zaffran et al., 2022), on streaming data with distribution shifts. FACI extends ACI by using a sliding window of conformity scores with size $w$ and learning rate $\eta$ for quantile updates. We test three scenarios: gradual shifts (linear transition over 200 timesteps), sudden shifts (instantaneous change), and recurring shifts (periodic changes every 400 timesteps). All methods use $\alpha = 0.1$ with 2000-sample streams including a 50-sample warm-up period excluded from metrics. Table 4 shows that Adaptive PBIS is the only method maintaining valid coverage ($\geq 88\%$) across all scenarios, achieving 90.3% average coverage. While ACI provides narrower intervals (width $= 10.51$), it suffers important failure under gradual shifts (84.2% coverage). Figure 2 illustrates this difference: during gradual shifts, ACI and FACI's coverage drops below 85%, violating the validity requirement. Adaptive PBIS's model selection mechanism, maintaining a posterior over 10 diverse models, enables robust adaptation where single-model approaches fail. Full temporal dynamics and detailed analysis are provided in Appendix B.3.

**Theoretical Validation.** We empirically validate that PBIS and PAC-Bayes achieve the theoretical $O(n^{-1/2})$ convergence rate for coverage error. Using 200 independent trials per sample size (ranging from 100 to 10,000), we fit the power law error $= c\,n^{-\beta}$ and estimate $\beta$ with bootstrap confidence intervals. Table 5 confirms both methods are statistically consistent with the theoretical rate, as their confidence intervals include $\beta = 0.5$. For quantile concentra-

tion, PBIS satisfies the Dvoretzky-Kiefer-Wolfowitz bound while PAC-Bayes shows a marginal violation (6.7% excess), expected for model averaging methods due to the Rademacher complexity of convex combinations. Figure 3 visualizes these theoretical properties. The parallel slopes in the left panel confirm both methods achieve the same convergence rate, while the right panel shows that PBIS's model selection strategy yields tighter prediction intervals without sacrificing coverage validity. Further analysis details are presented in Appendix B.4.

**Temperature Parameter $\lambda$ Analysis.** We first investigate the effect of the temperature parameter $\lambda$ on PBIS performance. We test fixed values $\lambda \in \{0.1, 0.5, 1.0, 2.0, 5.0, 10.0\}$ and adaptive schemes (sqrt, log, linear) across 7 datasets including 3 synthetic heteroscedastic datasets designed to reveal temperature effects. The results reveal two key insights. First, the temperature parameter correctly controls the exploration-exploitation tradeoff: selection entropy decreases monotonically from 1.884 ($\lambda = 0.1$) to 1.144 ($\lambda = 10.0$), with an entropy ratio of 1.41 confirming that lower temperatures lead to more exploratory selection (Table 6). Second, and more importantly, all configurations produce nearly identical mean widths across all datasets (1152.9) with coverage $0.914 \pm 0.038$, demonstrating that PBIS is remarkably robust to temperature choice when there exists a clearly superior model in the en-

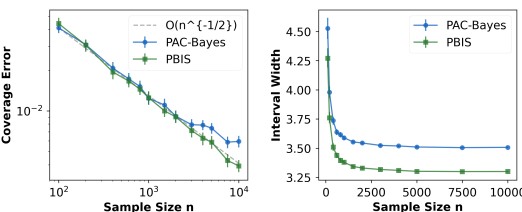

Figure 3: Convergence analysis. (Left) Coverage error convergence on log-log scale showing both methods achieve the theoretical rate. (Right) Interval width convergence indicating rapid stabilization with PBIS producing narrower intervals.

Table 5: Convergence rates and concentration properties

| Method | Convergence Rate ($\beta$) | | Concentration | |
| --- | --- | --- | --- | --- |
| | Coverage | 95% CI | Std Dev | DKW Bound |
| PAC-Bayes | 0.449 | [0.422, 0.507] | 0.0648 | ✗ |
| PBIS | 0.528 | [0.518, 0.541] | 0.0579 | ✓ |
| Theoretical | 0.500 | – | 0.0607 | – |

semble. This robustness holds even on synthetic heteroscedastic data where all configurations achieve width $6.96 \pm 0.44$. The adaptive sqrt scheme closely matches the theoretical optimal $\lambda = \sqrt{2\log(K)/n}$, providing a parameter-free default. We recommend $\lambda \in [1.0, 2.0]$ for practitioners seeking a fixed value. $\lambda$ is further analysed in Appendix C.1.

**Exploration Rate $\epsilon_n$.** We investigate the exploration-exploitation tradeoff in PBIS by testing exploration rates $\epsilon \in \{0.0, 0.01, 0.05, 0.1, 0.2, 0.5\}$ with five decay strategies (constant, square root, logarithmic, linear, exponential) across six datasets. To properly assess exploration benefits, we employ a diverse 10-model ensemble achieving average model disagreement CV = 0.261. All experiments use calibration size $n = 500$ with 10 independent trials per configuration. Note

Table 6: Selection entropy by configuration (lower = more deterministic).

| Fixed $\lambda$: | 0.1 | 0.5 | 1.0 | 2.0 | 5.0 | 10.0 |
| --- | --- | --- | --- | --- | --- | --- |
| **Entropy:** | 1.88 | 1.76 | 1.70 | 1.62 | 1.54 | 1.14 |

| **Adaptive:** | Sqrt | Log | Linear |
| --- | --- | --- | --- |
| **Entropy:** | 1.88 | 1.92 | 2.34 |

that pure exploitation ($\epsilon = 0$) achieves optimal performance with normalized width of 2.193 and average regret of 0.171, while maintaining valid coverage (89.9%). As shown in Figure 4, both width and regret increase monotonically with $\epsilon$, reaching 2.275 (+3.7%) and 0.219 (+40%) respectively at $\epsilon = 0.5$. Model selection diversity increases from 4.3 models at $\epsilon = 0$ to 9.8 models at $\epsilon = 0.5$, yet this diversity provides no performance benefit. The temperature-scaled posterior $\rho_i \propto \pi_i \exp(-\lambda Q_{1-\alpha}[s_i])$ already provides sufficient implicit exploration, making explicit $\epsilon$-greedy selection redundant. We recommend $\epsilon = 0$ as the default, simplifying deployment while maintaining optimal performance. Refer to Appendix C.2 for a detailed $\epsilon_n$ analysis.

**Prior Distribution.** We investigate the impact of prior distributions on PBIS performance across varying calibration sizes to understand when sophisticated priors provide practical benefits. We evaluated eight prior types (uniform, complexity-based, performance-based, maximum entropy, random, and three levels of misspecification) across calibration sizes $n \in \{50, 100, 200, 500\}$ using 6 datasets with 10 trials each. Table 7 shows that prior selection has limited practical impact on PBIS performance. While substantial width differences exist with small calibration sets ($n = 50$), all priors converge to similar performance at practical calibration sizes ($n \geq 200$). Surprisingly, the complexity-biased prior underperforms with limited data, suggesting aggressive regularization toward simple models can be counterproductive. PBIS maintains valid coverage (92% $\pm$ 1%) across all prior configurations, including severe misspecification (results in Appendix C.3). Thus, the uni-

Table 7: Prior impact diminishes with calibration size. Percentages show change relative to uniform baseline. Bold indicates best (lowest width) performers.

| Prior Type | Average Width by Calibration Size | | | |
|---|---|---|---|---|
| | $n = 50$ | $n = 100$ | $n = 200$ | $n = 500$ |
| Uniform | **1474** | **1290** | 1292 | 1501 |
| Complexity | 2516 (+70.7%) | 1627 (+26.1%) | 1355 (+4.9%) | **1489** (-0.8%) |
| Performance | 1686 (+14.3%) | 1406 (+8.9%) | 1270 (-1.7%) | 1573 (+4.8%) |
| Entropy | 1601 (+8.6%) | 1322 (+2.5%) | **1261** (-2.4%) | **1489** (-0.8%) |
| Coverage range: 0.909–0.926 (all maintain valid coverage $\geq 1$-$\alpha$) | | | | |

**Algorithm 2** Adaptive PBIS

**Require:** Initial posterior $\rho_0$, exploration rate $\epsilon_0$, learning rate $\eta$
1: **for** $t = 1, 2, \ldots$ **do**
2:     **if** $\text{random}() < \epsilon_t$ **then**
3:         $\theta_t \sim \rho_{t-1}$               ▷ Explore
4:     **else**
5:         $\theta_t = \text{mode}(\rho_{t-1})$      ▷ Exploit
6:     **end if**
7:     $C_t = \text{ConformalPredict}(s_{\theta_t}, x_t)$
8:     Observe $y_t$, update posterior based on performance
9:     $\epsilon_t = \epsilon_0 / \sqrt{t}$     ▷ Decay exploration
10: **end for**

form prior provides robust default performance across all conditions, eliminating the need for prior tuning in practical applications.

## 6 RELATED WORK AND DISCUSSION

Our work intersects PAC-Bayes theory (McAllester, 1999; Catoni, 2007), conformal prediction (Vovk et al., 2005; Romano et al., 2019), and adaptive inference under distribution shift (Gibbs & Candes, 2021; Zaffran et al., 2022). Sharma et al. (2023) proposed using PAC-Bayes to aggregate score functions in conformal prediction, assuming different scores excel in different regions. While intuitive, our theoretical analysis reveals why this aggregation approach fundamentally fails: the average-quantile divergence phenomenon causes systematic selection of suboptimal quantiles.

Recent work addresses distribution shift through various mechanisms. Gibbs & Candès (2024) introduced Adaptive Conformal Inference (ACI) with gradient-

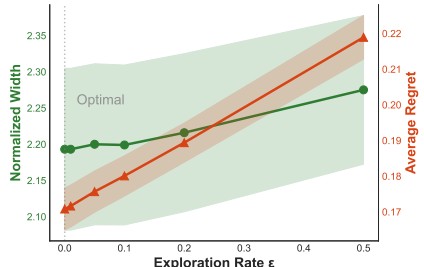

Figure 4: Exploration-exploitation tradeoff in PBIS. Both normalized width and average regret increase monotonically with exploration rate $\epsilon$. Pure exploitation ($\epsilon = 0$) achieves optimal performance. Shaded regions show 95% confidence intervals.

based quantile updates, while Zaffran et al. (2022) developed adaptive conformal predictions for time series using sliding windows. Gibbs & Candès (2024) extends ACI to arbitrary distribution shifts in online settings. Zhang et al. (2024) proposes Bayesian approaches for online conformal prediction. Compared to ACI and a sliding window variant method (FACI), our selection-based approach uniquely maintains valid coverage across all shift types by avoiding aggregation-induced inefficiency. Handling heteroscedasticity remains challenging for conformal prediction. Guha et al. (2024) converts regression to classification for better handling of heteroscedastic outputs, while Sebastián et al. (2024) combines heteroscedastic quantile regression with width-adaptive conformal inference. Dewolf et al. (2023) analyzes conditional validity under heteroscedasticity, showing how different conformal predictors relate to implicit distributional assumptions. Our work demonstrates that selection-based approaches naturally adapt to heteroscedasticity without requiring specialized scoring functions. Moreover, the tension between selection and aggregation extends beyond conformal prediction. While ensemble methods are popular in machine learning, recent work questions when aggregation helps versus hurts. Stanton et al. (2023) uses conformal prediction sets for Bayesian optimization, finding that careful selection often outperforms averaging. Similarly, our average-quantile divergence phenomenon suggests that selection may be broadly preferable when optimizing non-average functionals. Thus, we identified a fundamental incompatibility between PAC-Bayes aggregation and conformal prediction efficiency. The average-quantile divergence phenomenon shows that optimizing average scores can select models with poor quantile behavior, precisely what determines prediction set size. Our selection-based PBIS framework sidesteps this limitation while maintaining theoretical guarantees. The implications extend beyond conformal prediction. Any setting where tail behavior matters more than average performance—risk-sensitive RL, robust optimization, or extreme value prediction—may exhibit similar aggregation failures. Our results suggest that the intuitive appeal of aggregation must be carefully weighed against mathematical compatibility with the actual optimization objective.

## ETHICS STATEMENT

This work presents methodological advances in conformal prediction with no direct ethical concerns. All experiments use publicly available benchmark datasets without sensitive personal information. The proposed method improves uncertainty quantification in machine learning, which could enhance safety in deployed systems by providing more reliable prediction intervals.

## REPRODUCIBILITY STATEMENT

Code to reproduce all experiments is available at https://anonymous.4open.science/r/PBIS-9218/README.md. The implementation uses standard Python libraries (scikit-learn, numpy) with 27 public datasets from UCI, OpenML, and scikit-learn repositories. (single CPU, typically 5-10 minutes per dataset for complete evaluation). Theoretical proofs with complete derivations are provided in Appendix.

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
