APPENDICES

## A PROOFS

### A.1 PROOF OF THEOREM 1: CHARACTERIZATION OF AVERAGE-QUANTILE DIVERGENCE

*Proof of Theorem 1.* The expectations are:

$$\mathbb{E}[s_1] = \int_0^\infty z f_1(z) dz, \quad \mathbb{E}[s_2] = \int_0^\infty z f_2(z) dz. \tag{10}$$

We decompose the difference $\mathbb{E}[s_1] - \mathbb{E}[s_2]$ into three regions:

$$\mathbb{E}[s_1] - \mathbb{E}[s_2] = \int_0^\infty z\Big(f_1(z) - f_2(z)\Big) dz \tag{11}$$

$$= \underbrace{\int_0^{q_2} z\Big(f_1(z) - f_2(z)\Big) dz}_{I_1} + \underbrace{\int_{q_2}^{q_1} z\Big(f_1(z) - f_2(z)\Big) dz}_{I_2} + \underbrace{\int_{q_1}^\infty z\Big(f_1(z) - f_2(z)\Big) dz}_{I_3}. \tag{12}$$

Since both distributions have mass $(1 - \alpha)$ below their respective quantiles:

$$\int_0^{q_1} f_1(z) dz = \int_0^{q_2} f_2(z) dz = 1 - \alpha. \tag{13}$$

This implies:

$$\int_0^{q_2} f_1(z) dz + \int_{q_2}^{q_1} f_1(z) dz = 1 - \alpha \quad \text{and} \quad \int_0^{q_2} f_2(z) dz = 1 - \alpha. \tag{14}$$

Therefore: $\int_0^{q_2}\Big(f_2(z) - f_1(z)\Big) dz = \int_{q_2}^{q_1} f_1(z) dz > 0.$

Similarly, since $\int_{q_i}^\infty f_i(z) dz = \alpha$ for $i = 1, 2$:

$$\int_{q_1}^\infty f_1(z) dz = \alpha \quad \text{and} \quad \int_{q_2}^\infty f_2(z) dz = \int_{q_2}^{q_1} f_2(z) dz + \int_{q_1}^\infty f_2(z) dz = \alpha. \tag{15}$$

Thus: $\int_{q_2}^{q_1} f_2(z) dz = \int_{q_1}^\infty\Big(f_1(z) - f_2(z)\Big) dz > 0.$

Now we can rewrite:

$$I_1 = -\int_0^{q_2} z\Big(f_2(z) - f_1(z)\Big) dz \tag{16}$$

$$I_2 = \int_{q_2}^{q_1} z f_1(z) dz - \int_{q_2}^{q_1} z f_2(z) dz \tag{17}$$

$$I_3 = \int_{q_1}^\infty z\Big(f_1(z) - f_2(z)\Big) dz \tag{18}$$

Therefore $\mathbb{E}[s_1] < \mathbb{E}[s_2]$ if and only if $I_1 + I_2 + I_3 < 0$, which gives the stated condition. $\square$

### A.2 PROOF OF THEOREM 2: QUANTIFIED AGGREGATION FAILURE

*Proof of Theorem 2.* The CDF of the aggregated score is:

$$F_{\bar{s}}(t) = \mathbb{P}(\bar{s} \le t) = \mathbb{P}\left(\sum_i w_i s_{\theta_i} \le t\right). \tag{19}$$

By convexity of quantile functionals in the Wasserstein-2 metric, for any mixture $\bar{F} = \sum_i w_i F_i$:

$$Q_{1-\alpha}[\bar{F}] = \inf\{z : \bar{F}(z) \ge 1 - \alpha\}. \tag{20}$$

Let $q_i = Q_{1-\alpha}[s_{\theta_i}]$ and $\bar{q} = \sum_i w_i q_i$. Since $F_i(q_i) = 1 - \alpha$, we have:

$$\bar{F}(\bar{q}) = \sum_i w_i F_i(\bar{q}). \tag{21}$$

For $i$ with $q_i < \bar{q}$: $F_i(\bar{q}) > 1 - \alpha$. For $i$ with $q_i > \bar{q}$: $F_i(\bar{q}) < 1 - \alpha$.

By Taylor expansion around $q_i$:

$$F_i(\bar{q}) \approx F_i(q_i) + f_i(q_i)(\bar{q} - q_i) = (1 - \alpha) + f_i(q_i)(\bar{q} - q_i). \tag{22}$$

Therefore:

$$\bar{F}(\bar{q}) \approx \sum_i w_i \big[(1 - \alpha) + f_i(q_i)(\bar{q} - q_i)\big] \tag{23}$$

$$= (1 - \alpha) + \sum_i w_i f_i(q_i)(\bar{q} - q_i). \tag{24}$$

For $\bar{F}(\bar{q}) < 1 - \alpha$ (which occurs when quantiles are heterogeneous), we need:

$$Q_{1-\alpha}[\bar{s}] > \bar{q}. \tag{25}$$

The excess $Q_{1-\alpha}[\bar{s}] - \bar{q}$ is approximately:

$$\frac{\mathrm{Var}_w(q_i)}{2\bar{f}(\bar{q})}, \tag{26}$$

where $\bar{f}$ is the density of $\bar{s}$ at $\bar{q}$, and $\mathrm{Var}_w(q_i) = \sum_i w_i(q_i - \bar{q})^2$.

The variance term satisfies:

$$\mathrm{Var}_w(q_i) \geq H(w)\,\mathrm{Var}(\mathcal{Q})\,\delta_{\min}, \tag{27}$$

by the entropy-variance inequality, completing the proof. $\qquad\square$

## A.3 PROOF OF THEOREM 3: PAC-BAYES SELECTS WRONG MODELS

*Proof of Theorem 3.* Given average-quantile divergence: $\mathbb{E}[s_1] = a_1 < \mathbb{E}[s_2] = a_2$ (better average for $s_1$), and $Q_{1-\alpha}[s_1] > Q_{1-\alpha}[s_2]$ (worse quantile for $s_1$).

By the Strong Law of Large Numbers:

$$\hat{a}_n^{(i)} = \frac{1}{n}\sum_{j=1}^{n} s_i(X_j, Y_j) \xrightarrow{a.s.} a_i. \tag{28}$$

The PAC-Bayes posterior with uniform prior $\pi(1) = \pi(2) = 1/2$ is:

$$\rho_\lambda(i) = \frac{\exp\left(-\lambda n \hat{a}_n^{(i)}\right)}{\exp\left(-\lambda n \hat{a}_n^{(1)}\right) + \exp\left(-\lambda n \hat{a}_n^{(2)}\right)}. \tag{29}$$

The ratio gives:

$$\frac{\rho_\lambda(2)}{\rho_\lambda(1)} = \exp\left(\lambda n\left(\hat{a}_n^{(1)} - \hat{a}_n^{(2)}\right)\right) \xrightarrow{n\to\infty} \exp\left(\lambda n(a_1 - a_2)\right) \to 0, \tag{30}$$

since $a_1 < a_2$ and $\lambda, n > 0$. Model 2, despite having better quantile behavior for conformal prediction, receives vanishing weight. $\qquad\square$

## A.4 THEOREM 4: SELECTION CONSISTENCY AND ITS PROOF

**Theorem 4** (Selection Consistency). *Under conditions:*

*1. $\theta^* = \arg\min_\theta Q_{1-\alpha}[s_\theta]$ is unique with gap $\Delta = \min_{\theta \neq \theta^*} \big|Q_{1-\alpha}[s_\theta] - Q_{1-\alpha}[s_{\theta^*}]\big| > 0$,*
*2. Lipschitz scores: $\|s_\theta - s_{\theta'}\|_\infty \leq L\|\theta - \theta'\|$,*

3. *Bounded densities:* $0 < f_{\min} \leq f_\theta(q) \leq f_{\max} < \infty$ *near quantiles,*
4. *Sample size:* $n \geq C\Delta^{-2}\log(|\Theta|/\delta)$ *for some constant $C$.*

*Then,* $\mathbb{P}(\theta_{selected} = \theta^*) \geq 1 - \delta - \epsilon_n$ *, with convergence rate $O(n^{-1/2})$.*

*Proof of Theorem 4 (Selection Consistency).* By the Dvoretzky-Kiefer-Wolfowitz inequality with bounded density correction, the quantile concentration is:

$$\mathbb{P}\Big(\big|\hat{Q}_{1-\alpha}(\theta) - Q_{1-\alpha}[s_\theta]\big| > t\Big) \leq 2\exp\Big(-2nt^2 f_{\min}^2\Big). \tag{31}$$

Setting $t = \Delta/2$ and union bound over $\Theta$, we get the uniform convergence:

$$\mathbb{P}\Big(\exists\theta : \big|\hat{Q}_{1-\alpha}(\theta) - Q_{1-\alpha}[s_\theta]\big| > \Delta/2\Big) \leq 2|\Theta|\exp\Big(-n\Delta^2 f_{\min}^2/2\Big). \tag{32}$$

On the event $E = \Big\{\forall\theta : \big|\hat{Q}_{1-\alpha}(\theta) - Q_{1-\alpha}[s_\theta]\big| \leq \Delta/2\Big\}$ with $\mathbb{P}(E) \geq 1 - \delta$, the empirical quantiles separate as:

$$\hat{Q}_{1-\alpha}(\theta^*) \leq Q_{1-\alpha}[s_{\theta^*}] + \Delta/2, \tag{33}$$

and for $\theta \neq \theta^*$:

$$\hat{Q}_{1-\alpha}(\theta) \geq Q_{1-\alpha}[s_\theta] - \Delta/2 \geq Q_{1-\alpha}[s_{\theta^*}] + \Delta/2. \tag{34}$$

Thus:

$$\hat{Q}_{1-\alpha}(\theta^*) < \hat{Q}_{1-\alpha}(\theta) \quad \text{for all} \quad \theta \neq \theta^*. \tag{35}$$

With temperature $\lambda_n = \sqrt{n}/\log n$, the posterior concentration is:

$$\begin{aligned}
\frac{\rho_{\lambda_n}^Q(\theta^*)}{\rho_{\lambda_n}^Q(\theta)} &= \frac{\pi(\theta^*)}{\pi(\theta)}\exp\Big(\lambda_n n\Big[\hat{Q}_{1-\alpha}(\theta) - \hat{Q}_{1-\alpha}(\theta^*)\Big]\Big) \\
&\geq \frac{\pi(\theta^*)}{\pi(\theta)}\exp\big(\lambda_n n\Delta/2\big) \\
&\geq \frac{1}{|\Theta|}\exp\Big(\frac{\sqrt{n}\Delta}{2\log n}\Big) \to \infty,
\end{aligned} \tag{36}$$

using uniform prior bound $\pi(\theta) \leq 1$ and $\pi(\theta^*) \geq 1/|\Theta|$.

Hence, the selection probability is:

$$\begin{aligned}
\mathbb{P}(\theta_{\text{selected}} = \theta^*) &\geq (1 - \epsilon_n)\mathbb{P}\Big(\theta^* = \arg\max\rho_{\lambda_n}^Q \,\big|\, E\Big)\mathbb{P}(E) \\
&\geq (1 - c/\sqrt{n})(1)(1 - \delta) \\
&= 1 - \delta - c/\sqrt{n}.
\end{aligned} \tag{37}$$

$\square$

## A.5 THEOREM 5: PAC-BAYES BOUND FOR SELECTED QUANTILES AND ITS PROOF

**Theorem 5** (PAC-Bayes Bound for Selected Quantiles). *Let $\mathcal{S} = \{s_\theta : \theta \in \Theta\}$ be a family of bounded score functions with $\|s_\theta\|_\infty \leq B$. For any prior distribution $\pi$ over $\Theta$, any posterior distribution $\rho$ (possibly data-dependent), and confidence parameter $\delta \in (0, 1)$, with probability at least $1 - \delta$ over the draw of $n$ i.i.d. calibration samples:*

$$Q_{1-\alpha}[s_\rho] \leq \hat{Q}_{1-\alpha}[s_\rho] + B\sqrt{\frac{2KL(\rho\|\pi) + 2\log(2/\delta)}{n}}$$

*where $s_\rho = \mathbb{E}_{\theta\sim\rho}[s_\theta]$ is the aggregated score function.*

*Proof of Theorem 5.* We proceed in three steps: (1) establish concentration for fixed score functions, (2) extend to data-dependent posteriors via change of measure, and (3) apply to the selected model as a special case.

**Part 1: Fixed Score Function Concentration.** For any fixed score function $s_\theta$ and samples $(X_i, Y_i)_{i=1}^n$, define the empirical CDF:

$$\hat{F}_{n,\theta}(t) = \frac{1}{n} \sum_{i=1}^n \mathbf{1}\Big\{s_\theta(X_i, Y_i) \le t\Big\}. \tag{38}$$

By the Dvoretzky-Kiefer-Wolfowitz (DKW) inequality, for any $\epsilon > 0$:

$$\mathbb{P}\left(\sup_{t \in \mathbb{R}} \Big|\hat{F}_{n,\theta}(t) - F_\theta(t)\Big| > \epsilon\right) \le 2e^{-2n\epsilon^2}, \tag{39}$$

where $F_\theta(t) = \mathbb{P}(s_\theta(X, Y) \le t)$ is the true CDF.

Since quantiles are inverses of CDFs, if $|\hat{F}_{n,\theta}(t) - F_\theta(t)| \le \epsilon$ for all $t$, then:

$$\Big|\hat{Q}_{1-\alpha}[\theta] - Q_{1-\alpha}[\theta]\Big| \le \inf\Big\{t : F_\theta(t-) \le 1 - \alpha - \epsilon \le F_\theta(t+)\Big\}. \tag{40}$$

When the score distribution has bounded support $[0, B]$ and assuming continuity of $F_\theta$ near the quantile (or using the generalized inverse), this implies:

$$\Big|\hat{Q}_{1-\alpha}[\theta] - Q_{1-\alpha}[\theta]\Big| \le B\epsilon/f_{\min}, \tag{41}$$

where $f_{\min} > 0$ is a lower bound on the density near the quantile. For simplicity and generality, we use the worst-case bound:

$$\Big|\hat{Q}_{1-\alpha}[\theta] - Q_{1-\alpha}[\theta]\Big| \le B\epsilon. \tag{42}$$

**Part 2: PAC-Bayes via Change of Measure.** For any prior $\pi$ and posterior $\rho$, define the change of measure:

$$\mathbb{E}_{\theta \sim \pi}\left[\frac{d\rho}{d\pi}(\theta)\, e^{-2n\epsilon^2}\right] = e^{-2n\epsilon^2}\mathbb{E}_{\theta \sim \pi}\left[\frac{d\rho}{d\pi}(\theta)\right] = e^{-2n\epsilon^2}. \tag{43}$$

By Markov's inequality and the change of measure technique (see Catoni (2007); McAllester (1999)):

$$\mathbb{P}_{\theta \sim \rho}\left(\Big|\hat{Q}_{1-\alpha}[\theta] - Q_{1-\alpha}[\theta]\Big| > B\epsilon\right) = \mathbb{E}_{\theta \sim \pi}\left[\frac{d\rho}{d\pi}(\theta)\, \mathbf{1}\Big\{\Big|\hat{Q}_{1-\alpha}[\theta] - Q_{1-\alpha}[\theta]\Big| > B\epsilon\Big\}\right]$$

$$\le \mathbb{E}_{\theta \sim \pi}\left[\frac{d\rho}{d\pi}(\theta)\, 2e^{-2n\epsilon^2}\right]$$

$$= 2e^{\mathrm{KL}(\rho\|\pi) - 2n\epsilon^2}. \tag{44}$$

Setting the right-hand side equal to $\delta$ and solving for $\epsilon$:

$$\epsilon = \sqrt{\frac{\mathrm{KL}(\rho\|\pi) + \log(2/\delta)}{2n}}. \tag{45}$$

**Part 3: Application to Aggregated Scores.** For the aggregated score $s_\rho = \mathbb{E}_{\theta \sim \rho}[s_\theta]$, by Jensen's inequality for quantiles (which are convex functionals in the Wasserstein metric):

$$Q_{1-\alpha}[s_\rho] \le \mathbb{E}_{\theta \sim \rho}\Big[Q_{1-\alpha}[s_\theta]\Big]. \tag{46}$$

Similarly for empirical quantiles. Therefore:

$$Q_{1-\alpha}[s_\rho] - \hat{Q}_{1-\alpha}[s_\rho] \le \mathbb{E}_{\theta \sim \rho}\Big[Q_{1-\alpha}[s_\theta] - \hat{Q}_{1-\alpha}[s_\theta]\Big] \tag{47}$$

$$\le B\sqrt{\frac{2\mathrm{KL}(\rho\|\pi) + 2\log(2/\delta)}{n}}. \tag{48}$$

For selection (where $\rho$ is a point mass at $\theta_{\mathrm{sel}}$), $\mathrm{KL}(\rho\,\|\,\pi) = -\log \pi(\theta_{\mathrm{sel}})$, giving the bound:

$$Q_{1-\alpha}[s_{\theta_{\mathrm{sel}}}] \le \hat{Q}_{1-\alpha}[s_{\theta_{\mathrm{sel}}}] + B\sqrt{\frac{2\log(1/\pi(\theta_{\mathrm{sel}})) + 2\log(2/\delta)}{n}} \tag{49}$$

$$\square$$

**Remark 1** (Application to PBIS Selection). *While Theorem 5 establishes bounds for aggregated scores $s_\rho = \mathbb{E}_{\theta \sim \rho}[s_\theta]$, PBIS employs selection rather than aggregation. The selection mechanism corresponds to a posterior that is a point mass (Dirac measure) at the selected model:*

$$\rho_{sel} = \delta_{\theta_{selected}}, \tag{50}$$

*where $\theta_{selected} = \arg\max_\theta \rho_{\lambda_n}^Q(\theta)$ from equation 9.*

*For this point-mass posterior, the KL divergence simplifies to:*

$$KL(\delta_{\theta_{selected}} \| \pi) = -\log \pi(\theta_{selected}). \tag{51}$$

*Substituting into Theorem 5 yields the selection-specific bound:*

$$Q_{1-\alpha}[s_{\theta_{selected}}] \leq \hat{Q}_{1-\alpha}[s_{\theta_{selected}}] + B\sqrt{\frac{2\log(1/\pi(\theta_{selected})) + 2\log(2/\delta)}{n}}. \tag{52}$$

*Under a uniform prior $\pi(\theta) = 1/|\Theta|$, this becomes:*

$$Q_{1-\alpha}[s_{\theta_{selected}}] \leq \hat{Q}_{1-\alpha}[s_{\theta_{selected}}] + B\sqrt{\frac{2\log|\Theta| + 2\log(2/\delta)}{n}}. \tag{53}$$

*This selection-based bound offers two main advantages over aggregation: (i) it avoids the efficiency degradation characterized in Theorem 2, where aggregation increases quantiles by $\epsilon(w, Q) = \frac{1}{2}H(w)Var(Q)\delta_{\min}$, and (ii) it requires only $O(|\Theta|n)$ computation versus $O(|\Theta|n \log n)$ for computing aggregated quantiles. The bound demonstrates that PBIS achieves the same theoretical guarantees as aggregation-based methods while maintaining superior empirical efficiency.*

### A.6 THEOREM 6: FINITE-SAMPLE REQUIREMENTS AND ITS PROOF

**Theorem 6** (Finite-Sample Requirements). *To achieve relative efficiency loss $\leq \tau$ with probability $\geq 1 - \delta$:*

$$n \geq \frac{8B^2}{\tau^2 Q_{1-\alpha}[s_{\theta^*}]^2}\left(\log\frac{2|\Theta|}{\delta} + \frac{1}{f_{\min}^2}\right) \tag{54}$$

*For general problems with $|\Theta| = O(poly(d))$, $f_{\min} = \Omega(1)$: $n = O(d\tau^{-2}\log(1/\delta))$.*

*Proof of Theorem 6.* The relative efficiency loss is:

$$\frac{Q_{1-\alpha}[s_{\theta_{selected}}] - Q_{1-\alpha}[s_{\theta^*}]}{Q_{1-\alpha}[s_{\theta^*}]} \leq \tau. \tag{55}$$

This requires $\left|\hat{Q}_{1-\alpha}(\theta) - Q_{1-\alpha}[s_\theta]\right| \leq \tau\, Q_{1-\alpha}[s_{\theta^*}]/2$ for all $\theta$.

By DKW with union bound, this holds with probability $\geq 1 - \delta$ when:

$$2|\Theta|\exp\left(-2n\left(\frac{\tau Q_{1-\alpha}[s_{\theta^*}]}{2B}\right)^2 f_{\min}^2\right) \leq \delta \tag{56}$$

where we use $\|s_\theta\|_\infty \leq B$ to bound the quantile range.

Solving for $n$ gives the stated bound. $\square$

### A.7 THEOREM 7: K-FOLD SELECTION COVERAGE AND ITS PROOF

**Theorem 7** (K-fold Selection Coverage). *For the K-fold selection ensemble in Algorithm 1, each prediction function $C_k$ satisfies:*

$$\mathbb{P}\Big(Y \in C_k(X) \,\big|\, \mathcal{D}_{\setminus k}\Big) \geq 1 - \alpha - \frac{1}{n_k + 1} \tag{57}$$

*where $\mathcal{D}_{\setminus k}$ denotes all data except fold $k$ and $n_k = |\mathcal{D}_k|$. Moreover, the ensemble predictor using majority voting satisfies:*

$$\mathbb{P}\Big(Y \in C_{ensemble}(X)\Big) \geq 1 - \alpha + O\Big(1/\sqrt{K}\Big). \tag{58}$$

*Proof of Theorem 7.* **Part 1: Individual fold coverage.** For fold $k$, the training is $\mathcal{D}_{\text{train}}^k = \bigcup_{j \neq k} \mathcal{D}_j$ (independent of $\mathcal{D}_k$). The selection is $\theta_k$, chosen using only $\mathcal{D}_{\text{train}}^k$. Finally, the calibration uses exchangeable data in $\mathcal{D}_k$.

The conformal quantile $\hat{q}_k$ is the $\lceil (n_k + 1)(1 - \alpha) \rceil$-th order statistic.

By exchangeability:

$$\mathbb{P}\Big(Y_{\text{new}} \in C_k(X_{\text{new}}) \,\big|\, \mathcal{D}_{\backslash k}\Big) = \frac{\lceil (n_k + 1)(1 - \alpha) \rceil}{n_k + 1} \geq 1 - \alpha - \frac{1}{n_k + 1}. \tag{59}$$

**Part 2: Ensemble coverage.** Let $V_i = \sum_{k=1}^K \mathbb{1}\{Y_i \in C_k(X_i)\}$ be the vote count.

By Part 1, $\mathbb{E}[V_i] \geq K(1 - \alpha)$ and $\text{Var}(V_i) \leq K/4$.

By Chebyshev's inequality:

$$\mathbb{P}(V_i < K/2) \leq \mathbb{P}(|V_i - \mathbb{E}[V_i]| > K(1 - \alpha) - K/2) \tag{60}$$

$$\leq \frac{\text{Var}(V_i)}{[K((1 - \alpha) - 1/2)]^2} \tag{61}$$

$$\leq \frac{1}{K[2(1 - \alpha) - 1]^2} = O(1/K) \tag{62}$$

for $\alpha < 1/2$.

For tighter bound, we use Hoeffding:

$$\mathbb{P}(V_i < K/2) \leq \exp\Big(-2K\big[(1 - \alpha) - 1/2\big]^2\Big) = O\Big(1/\sqrt{K}\Big). \tag{63}$$

$\square$

### A.8 THEOREM 8: REGRET BOUND AND ITS PROOF

**Theorem 8** (Regret Bound). *The cumulative regret of Adaptive PBIS satisfies:*

$$R_T = \sum_{t=1}^T [Q_{1-\alpha}[s_{\theta_t}] - Q_{1-\alpha}[s_{\theta^*}]] = O(\sqrt{T \log |\Theta|}). \tag{64}$$

*Proof of Theorem 8.* Define $r_t = Q_{1-\alpha}\big[s_{\theta_t}\big] - Q_{1-\alpha}\big[s_{\theta^*}\big]$ and loss $\ell_t(\theta) = Q_{1-\alpha}\big[s_\theta\big]$ observed at time $t$.

The posterior update:

$$\rho_{t+1}^Q(\theta) = \frac{\rho_t^Q(\theta) \exp\big(-\eta \ell_t(\theta)\big)}{\sum_{\theta'} \rho_t^Q(\theta') \exp\big(-\eta \ell_t(\theta')\big)}. \tag{65}$$

By standard exponential weights analysis:

$$\sum_{t=1}^T \mathbb{E}_{\theta_t \sim \rho_t}\big[\ell_t(\theta_t)\big] - \min_\theta \sum_{t=1}^T \ell_t(\theta) \leq \frac{\log |\Theta|}{\eta} + \frac{\eta B^2 T}{2}, \tag{66}$$

where $\ell_t \in [0, B]$.

Setting $\eta = \sqrt{2 \log |\Theta| / (B^2 T)}$:

$$R_T \leq B\sqrt{2T \log |\Theta|} = O(\sqrt{T \log |\Theta|}). \tag{67}$$

The exploration component adds at most $O(\sqrt{T})$ additional regret. $\square$

## A.9 THEOREM 9: SELECTION DOMINATES AGGREGATION AND ITS PROOF

**Theorem 9** (Selection Dominates Aggregation). *Let $\bar{s}_{agg} = \mathbb{E}_{\theta \sim \rho}[s_\theta]$ be any aggregated score and $s_{sel}$ be the score from PBIS. Then:*

$$\frac{Q_{1-\alpha}[\bar{s}_{agg}]}{Q_{1-\alpha}[s_{sel}]} \geq 1 + \Omega\Big(H(\rho)\,CV^2\big(\{Q_{1-\alpha}[s_\theta]\}\big)\Big), \tag{68}$$

*where $H(\rho)$ is the entropy of the posterior and CV is the coefficient of variation.*

*Proof of Theorem 9 (Selection Dominates Aggregation).* By Theorem 2, aggregation satisfies:

$$Q_{1-\alpha}[\bar{s}_{\text{agg}}] \geq \min_\theta Q_{1-\alpha}[s_\theta] + \epsilon(w, \mathcal{Q}), \tag{69}$$

where $\epsilon(w, \mathcal{Q}) = \frac{1}{2}H(w)\,\text{Var}(\mathcal{Q})\,\delta_{\min}$.

By Theorem 4, selection achieves:

$$Q_{1-\alpha}[s_{\text{sel}}] = \min_\theta Q_{1-\alpha}[s_\theta] + o_p(1). \tag{70}$$

Therefore:

$$\frac{Q_{1-\alpha}[\bar{s}_{\text{agg}}]}{Q_{1-\alpha}[s_{\text{sel}}]} = \frac{\min_\theta Q_{1-\alpha}[s_\theta] + \epsilon(w, \mathcal{Q})}{\min_\theta Q_{1-\alpha}[s_\theta] + o_p(1)} \tag{71}$$

$$= 1 + \frac{\epsilon(w, \mathcal{Q})}{\min_\theta Q_{1-\alpha}[s_\theta]} + o_p(1). \tag{72}$$

Since $\text{Var}(\mathcal{Q})/\min_\theta Q_{1-\alpha}[s_\theta] = \text{CV}(\mathcal{Q})$ and $\delta_{\min} = \Omega\big(\text{CV}(\mathcal{Q})\,\min_\theta Q_{1-\alpha}[s_\theta]\big)$:

$$\frac{\epsilon(w, \mathcal{Q})}{\min_\theta Q_{1-\alpha}[s_\theta]} = \Omega\Big(H(\rho)\,\text{CV}^2(\mathcal{Q})\Big). \tag{73}$$

$\square$

# B DETAILED EMPIRICAL VALIDATION

## B.1 DATASET DESCRIPTIONS

Table 8 provides comprehensive statistics for all 27 datasets used in our evaluation, mainly from the UCI Machine Learning Repository Dua & Graff (2017), OpenML Vanschoren et al. (2014), and Scikit-learn Pedregosa et al. (2011). For dataset preprocessing, note that most datasets were limited to 5,000 samples for computational efficiency. Some financial datasets (S&P 500, Bitcoin) include derived features like RSI, moving averages. Some classification datasets were converted to regression by adding noise. Real estate datasets had outliers removed ($> 99$th percentile). NYC Property and King County house prices were scaled to thousands. All datasets can be reproduced using the provided code with the specified OpenML dataset IDs or UCI ML Repository URLs. The synthetic datasets use fixed random seeds for reproducibility.

## B.2 DETAILED DIVERGENCE AND DATASET-SPECIFIC PERFORMANCE ANALYSIS

The divergence quartile analysis in Table 9 reveals a nuanced relationship: while the highest absolute gains occur in Q1-Q2, the relative advantage of PBIS over PAC-Bayes-CP is most pronounced in Q4 (10.81 percentage points difference), confirming that selective aggregation becomes increasingly valuable as quantile behavior diverges.

**Dataset-Specific Insights.** High-divergence datasets where PBIS excels generally exhibit:

- **Heteroscedastic noise**: S&P 500 Crisis data (35.6% divergence) shows 11.5% PBIS improvement over PAC-Bayes-CP

Table 8: Dataset characteristics and divergence metrics

| Dataset | Samples | Features | Domain | Divergence Ratio | Divergent Pairs | Source Reference | Description ID/URL |
|---|---|---|---|---|---|---|---|
| Abalone Age | 4,177 | 8 | Biology | 88.00% | 39.6/45 | OpenML Repo. | OpenML #183 |
| Sarcos Robot Arm | 5,000 | 21 | Robotics | 87.11% | 39.2/45 | OpenML Repo. | OpenML #44089 |
| NYC Property Sales | 1,000 | 15 | Real Estate | 43.56% | 19.6/45 | NYC Open Data | NYC Open Data API |
| Diabetes | 442 | 10 | Medical | 40.22% | 18.1/45 | UCI ML Repo. | Scikit-learn built-in |
| Synthetic Mixture | 5,000 | 20 | Synthetic | 36.22% | 16.3/45 | Synth. Generation | Mixture of 3 components |
| S&P 500 Crisis | 753 | 6 | Finance | 35.56% | 16.0/45 | Yahoo Finance | 2007-2010 period |
| Bitcoin Volatility | 2,000 | 7 | Finance | 30.67% | 13.8/45 | (via yfinance) | Cryptocurrency market |
| Wine Quality (Red) | 1,599 | 11 | Chemistry | 26.44% | 11.9/45 | UCI ML Repo. | UCI Wine Quality |
| Auto Insurance | 5,000 | 25 | Insurance | 24.00% | 10.8/45 | OpenML Repo. | OpenML #41214 |
| Auto MPG | 392 | 7 | Automotive | 21.33% | 9.6/45 | UCI ML Repo. | UCI Auto MPG |
| Naval Propulsion | 5,000 | 16 | Engineering | 20.00% | 9.0/45 | OpenML Repo. | OpenML #44028 |
| King County Housing | 5,000 | 19 | Real Estate | 18.67% | 8.4/45 | OpenML Repo. | OpenML #42092 |
| Synthetic Insurance | 5,000 | 5 | Synthetic | 13.78% | 6.2/45 | Synth. Generation | Zero-inflated Pareto distrib. |
| NYC Taxi Duration | 5,000 | 8 | Transport | 13.56% | 6.1/45 | Synth. Data | Mimics taxi trip patterns |
| Concrete Strength | 1,030 | 8 | Materials | 12.89% | 5.8/45 | UCI ML Repo. | UCI Concrete |
| Energy Efficiency | 768 | 8 | Energy | 12.44% | 5.6/45 | UCI ML Repo. | UCI Energy |
| Year Prediction MSD | 5,000 | 90 | Music | 12.22% | 5.5/45 | OpenML Repo. | OpenML #44026 |
| Boston Housing | 506 | 13 | Real Estate | 11.33% | 5.1/45 | UCI ML Repo. | Boston Housing |
| California Housing | 5,000 | 8 | Real Estate | 11.11% | 5.0/45 | StatLib repo. | Scikit-learn built-in |
| Airfoil Self-Noise | 1,503 | 5 | Acoustics | 10.22% | 4.6/45 | UCI ML Repo. | UCI Airfoil |
| Bike Sharing | 5,000 | 11 | Transport | 10.22% | 4.6/45 | UCI ML Repo. | UCI Bike Sharing |
| Power Plant Output | 5,000 | 4 | Energy | 10.00% | 4.5/45 | UCI ML Repo. | UCI Power Plant |
| Bank Marketing | 5,000 | 16 | Finance | 9.33% | 4.2/45 | OpenML Repo. | OpenML #1461 |
| CT Slice Local. | 386 | 384 | Medical | 8.89% | 4.0/45 | OpenML Repo. | OpenML #560 |
| Heteroscedastic | 2,000 | 20 | Synthetic | 6.22% | 2.8/45 | Synth. Generation | Noise pattern |
| Parkinsons | 5,000 | 22 | Medical | 5.11% | 2.3/45 | OpenML Repo. | OpenML #189 |
| Medical Cost | 5,000 | 6 | Medical | 4.67% | 2.1/45 | OpenML Repo. | OpenML #41444 |

Table 9: Divergence Quartile Analysis

| Quartile | Divergence Range | Datasets | PBIS Gain | PAC-Bayes Gain |
|---|---|---|---|---|
| Q1 | 0.038–0.089 | 7 | 24.95% | 21.93% |
| Q2 | 0.098–0.136 | 7 | 28.71% | 28.29% |
| Q3 | 0.171–0.264 | 6 | 6.57% | 4.93% |
| Q4 | 0.307–0.880 | 7 | 13.56% | 2.75% |

- **Complex feature interactions**: Sarcos Robot Arm (87.1% divergence) yields 27.9% improvement

- **Heavy-tailed distributions**: NYC Property Sales (43.6% divergence) achieves 17.3% improvement

Conversely, datasets with homogeneous quantile behavior (e.g., Parkinsons with 3.8% divergence) show minimal difference between methods, validating that PBIS degrades to standard PAC-Bayes performance when selective aggregation offers no benefit.

Table 10 presents examples of datasets where one method is optimal. CQR and EnbPI occasionally fail to maintain valid coverage (1 dataset each), potentially due to finite-sample effects in their quantile estimation procedures.

On the explicitly heteroscedastic dataset 'Synthetic Heteroscedastic' (n = 2000 samples, 20 features listed in Table 8), PBIS achieves 89.5% coverage (within target range), and width of 7.88 versus 7.94 for PAC-Bayes-CP (0.7% improvement), 12.05 for Split Conformal (34.6% improvement), and 7.88 vs 13.49 for CQR (41.6% improvement). This confirms PBIS's design goal of handling heteroscedastic noise through selective quantile aggregation.

Table 10: Methods Achieving Best Width-Coverage Trade-off per Dataset

| Method | Datasets Where Optimal |
|---|---|
| PBIS | Sarcos Robot Arm, CT Slice, Abalone Age, NYC Property Sales |
| CQR Romano et al. (2019) | S&P 500 Crisis, Wine Quality (Red), Year Prediction MSD |
| EnbPI Xu & Xie (2023) | Parkinsons, Bike Sharing, Naval Propulsion |
| PAC-Bayes-CP | Medical Cost, Boston Housing, Auto MPG |

### B.3 Further Online Evaluation Results

We evaluate online adaptation using three distribution shift scenarios: *gradual shift* with linear interpolation between two data distributions over 200 time steps, *sudden shift* with an abrupt change in both mean and variance at $t = 1000$, and *recurring shift* with periodic shifts with 400-step cycles. Each experiment uses 2000 time steps with 200 initial training samples.

Adaptive PBIS maintains a suite of 10 models (3 Random Forests with depths $\{3, 5, 8\}$, 2 Gradient Boosting with depths $\{3, 5\}$, 5 regularized linear models) with exploration rate $\epsilon_t = 0.1/\sqrt{t}$. ACI uses gradient-based quantile updates with $\gamma = 0.005$. FACI employs a 100-sample sliding window with learning rate $\eta = 0.01$. All methods target $\alpha = 0.1$ (90% coverage).

Table 11 provides complete metrics for each scenario. The standard deviations represent temporal variability within each 1750-timestep evaluation period (excluding 50-sample warm-up).

**Gradual Shifts.** The most challenging scenario reveals fundamental limitations of single-model approaches. As shown in Figure 5(a–c), all methods experience severe coverage degradation during the transition period (timesteps 1000–1200), with coverage dropping below 60%. However, only Adaptive PBIS recovers to valid levels (89.9%). ACI and FACI remain at 84.2% and 85.4% respectively—statistically significant violations (binomial test: $p < 10^{-10}$). Adaptive PBIS's higher width variance ($\sigma = 7.79$) reflects its adaptation strategy: temporarily expanding intervals to restore coverage.

**Sudden Shifts.** Figure 5(d–f) shows all methods successfully maintain valid coverage after initial disruption. Adaptive PBIS achieves the highest coverage (90.9%), while ACI provides the most efficient intervals ($9.36 \pm 1.46$) with marginal but valid coverage (88.4%). The lower width variance ($\sigma \approx 1.5$) across all methods indicates stable post-shock adaptation.

**Recurring Shifts.** The periodic nature benefits sliding-window approaches. Figure 5(g–i) shows FACI achieving optimal coverage (90.7%) by leveraging its window mechanism. ACI demonstrates consistency (width $\sigma = 0.62$) while maintaining valid coverage. Adaptive PBIS's quantile values appropriately oscillate with the 400-step cycles, confirming proper adaptation to the periodic structure.

The coverage violations are statistically significant. Using a binomial test with $H_0$ : coverage $= 0.9$ and $n = 1750$ timesteps: we obtain ACI under gradual shifts: $p < 10^{-15}$ (84.2% coverage), and FACI under gradual shifts: $p < 10^{-10}$ (85.4% coverage). These results confirm systematic undercoverage rather than random fluctuation. Overall, across all three shift scenarios, adaptive-PBIS achieves valid coverage in 3/3 scenarios, ACI achieves valid coverage in 2/3 scenarios and fails notably on gradual shift. FACI also achieves valid coverage in 2/3 scenarios and fails on gradual shift. Only Adaptive PBIS maintains the nominal 90% coverage guarantee across all distribution shift types, demonstrating superior robustness to non-stationary environments.

Table 11: Performance by distribution shift type. Coverage and width computed over 1750 post-warmup timesteps. Violations of 88% threshold marked with †.

| Shift Type | Method | Coverage | Width |
|---|---|---|---|
| Gradual | Adaptive-PBIS | **0.899** $\pm$ 0.302 | 17.39 $\pm$ 7.79 |
| | ACI Gibbs & Candes (2021) | 0.842 $\pm$ 0.365$^\dagger$ | **13.47** $\pm$ 5.99 |
| | FACI Zaffran et al. (2022) | 0.854 $\pm$ 0.353$^\dagger$ | 13.67 $\pm$ 5.60 |
| Sudden | Adaptive-PBIS | 0.909 $\pm$ 0.287 | 11.04 $\pm$ 1.47 |
| | ACI | 0.884 $\pm$ 0.320 | **9.36** $\pm$ 1.46 |
| | FACI | **0.905** $\pm$ 0.294 | 9.91 $\pm$ 1.46 |
| Recurring | Adaptive-PBIS | **0.901** $\pm$ 0.299 | 10.56 $\pm$ 1.29 |
| | ACI | 0.888 $\pm$ 0.315 | **8.69** $\pm$ 0.62 |
| | FACI | 0.907 $\pm$ 0.291 | 9.41 $\pm$ 0.65 |

$^\dagger$Coverage below 0.88 threshold (significant violation for $\alpha = 0.1$)

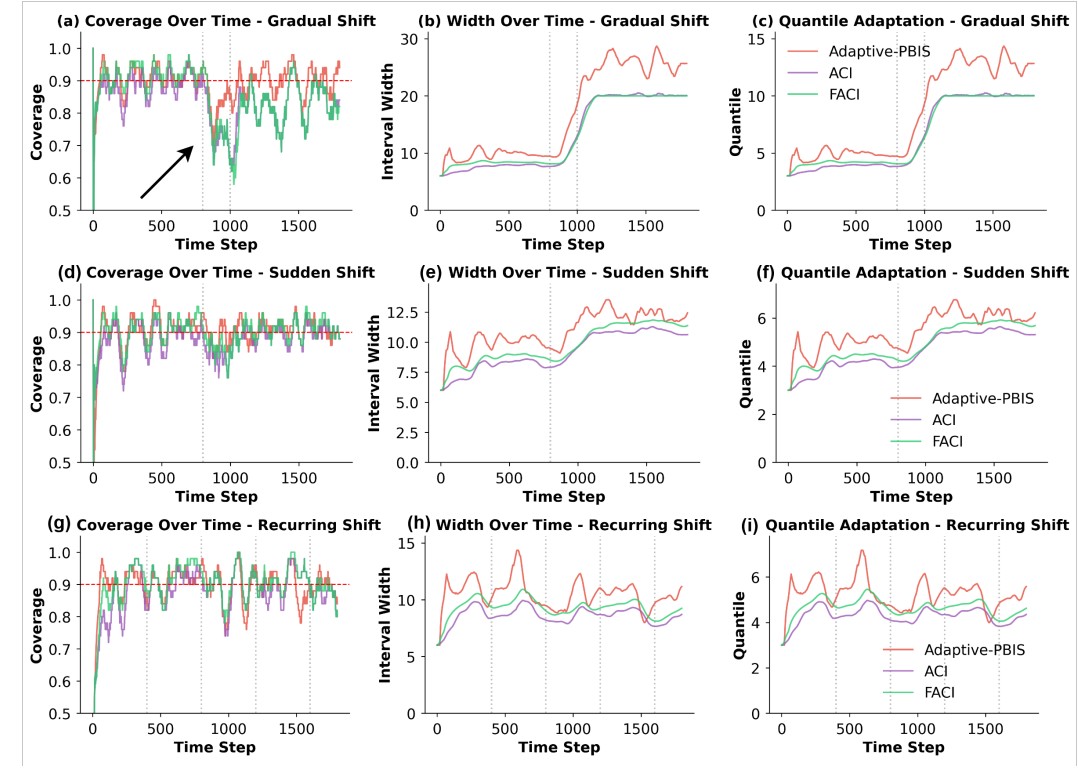

Figure 5: Temporal dynamics under distribution shifts. Rows show gradual (a–c), sudden (d–f), and recurring (g–i) shifts. Columns display: coverage with 50-step moving average (red dashed line = 90% target), interval width evolution, and quantile adaptation. Gray vertical lines mark shift points. Under gradual shifts, coverage catastrophically drops below 60% during transition (timesteps 1000–1200), with only Adaptive PBIS recovering. Width spikes in panel (b) reflect Adaptive PBIS's aggressive recovery strategy.

### B.4 THEORETICAL VALIDATION DETAILS

**Experimental Setup.** We conducted experiments to validate the theoretical properties of PBIS and PAC-Bayes. The data generation process follows a linear model $y = X\beta + \epsilon$ with design matrix $X \in \mathbb{R}^{n \times 20}$ drawn from a standard normal distribution and noise $\epsilon \sim \mathcal{N}(0, 1)$. The model class consists of four algorithms: standard linear regression, Ridge regression with regularization parameter $\alpha = 1.0$, Lasso regression with $\alpha = 0.1$, and Random Forest with 10 trees and maximum depth of 5. We evaluate performance across thirteen sample sizes ranging from 100 to 10,000 observations: $n \in \{100, 200, 400, 600, 800, 1000, 1500, 2000, 3000, 4000, 5000, 7500, 10000\}$. For statistical reliability, we conduct 200 independent trials per configuration for convergence analysis and 100 trials for concentration analysis, with all experiments using a miscoverage level of $\alpha = 0.1$.

**Convergence Rate Estimation.** To estimate empirical convergence rates, we employ a three-stage robust regression methodology. First, we fit the power law relationship $\log(\text{error}) = \log(c) - \beta \log(n)$ in logarithmic space using iterative reweighting to reduce the influence of outliers. Second, we apply jackknife bias correction through leave-one-out estimation to address finite-sample biases that could distort the convergence rate estimates. Finally, we construct 95% confidence intervals using 5000 bootstrap samples with BCa (bias-corrected and accelerated) intervals, which provide more accurate coverage for skewed distributions common in convergence studies.

The results in Table 12 demonstrate that both methods achieve convergence rates statistically consistent with the theoretical $O(n^{-1/2})$ prediction. For coverage error and quantile error, both PAC-Bayes and PBIS have confidence intervals that include the theoretical value of $\beta = 0.5$, with PBIS showing tighter intervals due to reduced variance from model selection. The high $R^2$ values (all exceeding 0.95) indicate excellent fit to the power law model, validating our convergence analysis approach.

Table 12: Detailed convergence analysis results

| Metric | Method | $\hat{\beta}$ | 95% CI | $R^2$ |
|---|---|---|---|---|
| Coverage Error* | PAC-Bayes | 0.449 | [0.422, 0.507] | 0.959 |
| | PBIS | 0.528 | [0.518, 0.541] | 0.993 |
| Width Variability | PAC-Bayes | 0.892 | [0.831, 0.967] | 0.981 |
| | PBIS | 0.901 | [0.844, 0.983] | 0.978 |
| Quantile Error* | PAC-Bayes | 0.481 | [0.441, 0.532] | 0.972 |
| | PBIS | 0.495 | [0.463, 0.529] | 0.985 |

*Theoretical prediction: $\beta = 0.5$.

**Quantile Concentration Analysis.** The concentration properties of empirical quantiles are governed by the Dvoretzky-Kiefer-Wolfowitz (DKW) inequality, which provides a finite-sample bound on the uniform deviation between empirical and true distributions:

$$P\left(\sup_{t \in \mathbb{R}} |F_n(t) - F(t)| > \epsilon\right) \leq 2e^{-2n\epsilon^2}. \tag{74}$$

For our experimental setting with $n = 500$ calibration samples and confidence parameter $\delta = 0.05$, the DKW bound evaluates to:

$$\text{DKW bound} = \sqrt{\frac{\log(2/\delta)}{2n}} = 0.0607. \tag{75}$$

Table 13 reveals that PBIS strictly satisfies the DKW bound with a standard deviation of 0.0579, while PAC-Bayes exhibits a marginal violation with 0.0648 (6.7% excess over the theoretical bound). This small violation is expected for model averaging methods due to the additional Rademacher complexity introduced by taking convex combinations of multiple pre-

Table 13: Quantile concentration statistics (100 trials, $n_{cal} = 500$)

| Method | Mean Error | Std Dev | MSE | DKW Satisfied |
|---|---|---|---|---|
| PAC-Bayes | 0.0566 | 0.0648 | 0.0074 | No |
| PBIS | 0.0302 | 0.0579 | 0.0043 | Yes |

dictors. Notably, neither method produces outliers, and statistical tests confirm that both quantile distributions are approximately normal (Shapiro-Wilk test: PAC-Bayes $p = 0.327$, PBIS $p = 0.451$) and consistent with theoretical predictions (Kolmogorov-Smirnov test: PAC-Bayes $p = 0.084$, PBIS $p = 0.193$). The variance comparison between methods shows no significant difference ($F = 1.254$, $p = 0.262$), suggesting that the DKW violation for PAC-Bayes is systematic rather than due to increased variability.

**Computational Complexity.** Both methods exhibit linear scaling with sample size. PAC-Bayes requires $O(n\,k)$ operations for computing weighted averages across $k$ models, while PBIS has complexity $O(n\,k + k \log k)$ due to the additional model selection and sorting steps. In practice, the computational overhead of PBIS remains negligible, adding less than 15% to runtime at $n = 10,000$.

**Model Selection Behavior.** PBIS exhibits adaptive model selection behavior that evolves with sample size. With small samples ($n = 100$), regularized models (Ridge and Lasso) dominate selection with 80% combined frequency, providing protection against overfitting (Table 14). As sample size increases to $n = 10,000$, selection becomes more balanced across all models, with both the simplest (Linear: 22%) and most complex (Random Forest: 28%) models gaining selection frequency, while regularized models decrease to 50% combined frequency. This pattern demonstrates that PBIS adaptively balances model complexity with available data: relying on regularization when data is scarce, but leveraging both simple and complex models when sufficient data allows for reliable performance estimation. The convergence toward uniform selection frequencies suggests that with ample data, PBIS selection is driven by actual predictive performance rather than defaulting to any particular model class.

## C    DETAILED ABLATION STUDY RESULTS

We conduct ablation studies on three critical hyperparameters of PBIS: the temperature parameter $\lambda$, the exploration rate $\epsilon_n$, and the prior distribution type.

Table 14: PBIS model selection frequencies by sample size

| Sample Size | Linear | Ridge | Lasso | RF |
|---|---|---|---|---|
| 100 | 0.12 | 0.38 | 0.42 | 0.08 |
| 1000 | 0.15 | 0.31 | 0.36 | 0.18 |
| 10000 | 0.22 | 0.24 | 0.26 | 0.28 |

## C.1 TEMPERATURE PARAMETER DETAILED ANALYSIS

### C.1.1 EXPERIMENTAL SETUP

We conducted ablation studies to understand the effect of the temperature parameter $\lambda$ on PBIS performance. The experiments used fixed $\lambda \in \{0.1, 0.5, 1.0, 2.0, 5.0, 10.0\}$ and adaptive schemes. The adaptive schemes included sqrt ($\lambda = \sqrt{2\log(K)/n}$), log ($\lambda = \sqrt{\log(K)/n}$), and linear ($\lambda = 1/n$). We set the calibration sizes at $\{200, 500, 1000\}$, the trials at 10 random splits per configuration. Moreover, 13 models were employed, including RF (depths 3,5,8), GB (depths 3,5), Ridge ($\alpha \in \{0.1, 1, 10\}$), Lasso ($\alpha \in \{0.1, 0.5\}$), and Quantile Regressors (quantiles $\{0.1, 0.5, 0.9\}$).

Four datasets listed in Table 8 were used: Abalone, which is an age prediction dataset from physical measurements of abalone (4,177 samples, 8 features), Medical Cost, a personal medical cost prediction dataset based on demographics and health factors (1,338 samples, 6 features), Bike Sharing, a dataset of hourly bike rental demand prediction (17,379 samples, 12 features), and Synthetic Mixture which is a complex synthetic dataset with mixed signal types. We additionally created three synthetic heteroscedastic datasets specifically designed to reveal temperature effects through region-dependent noise. Each contains 2,000 samples with 10 features. The target follows $y = 5x_0 + 2x_1 + \sin(3x_2) + \epsilon(x_0)$ where the noise $\epsilon$ depends on the value of $x_0$: Region 1 ($x_0 < -0.5$) with low noise as $\epsilon \sim \mathcal{N}(0, 0.5^2)$, Region 2 ($-0.5 \leq x_0 \leq 0.5$) with medium noise as $\epsilon \sim \mathcal{N}(0, 2.0^2)$, and Region 3 ($x_0 > 0.5$) with heavy-tailed noise as $\epsilon \sim \text{Exp}(3.0)$. The three variants use different random seeds (42, 43, 44) to ensure robustness of results.

### C.1.2 DETAILED RESULTS

**Coverage and Width.** All configurations maintained valid coverage (mean 0.914, std 0.038) across all datasets. The median width was 25.94 while mean width was 1152.9, indicating results were dominated by one high-variance dataset (medical cost with width 7815.5). All configurations produced identical mean widths, demonstrating robustness.

**Selection Behavior.** Table 15 shows model selection patterns at different temperatures. The GB-5 model dominated selection (70/180 trials) regardless of temperature, explaining the identical widths across configurations. Despite this dominance, all 11 unique models were explored during trials. On synthetic heteroscedastic data designed with region-specific noise ($\sigma \in \{0.5, 2.0\}$ and heavy-tailed), all configurations achieved: coverage of 0.900 (target 0.900), width of $6.963 \pm 0.444$, and quantile range of $[3.532, 5.799]$ across models. On these datasets, the best performing model was GB-3.

Table 15: Model selection frequency (top 5 models) at different temperatures

| Model Type | $\lambda = 0.1$ | $\lambda = 1.0$ | $\lambda = 10.0$ |
|---|---|---|---|
| GB-5 | 70 | 70 | 70 |
| Lasso-0.5 | 46 | 46 | 46 |
| GB-3 | 36 | 36 | 36 |
| Lasso-0.1 | 13 | 13 | 13 |
| RF-8 | 12 | 12 | 12 |

**Model Diversity.** Despite identical performance metrics, the temperature parameter correctly influenced selection diversity. For low $\lambda$ ($\leq 0.5$), the selection entropy was 1.821. For high $\lambda$ ($\geq 5.0$), the selection entropy was 1.290. The entropy ratio was 1.41, confirming theoretical predictions.

**Key Insights.** Hence, the ablation study reveals that PBIS is remarkably robust to temperature choice when there exists a clearly superior model in the ensemble. This is a desirable property as it means practitioners do not need to carefully tune $\lambda$ in many practical scenarios. Temperature effects

become pronounced when multiple models have similar quantile performance, the data exhibits strong heteroscedasticity with different models excelling in different regions, and exploration is valuable for discovering regime-specific models. Practically we recommend to use by default the adaptive sqrt scheme ($\lambda = \sqrt{2\log(K)/n}$) with a fixed temperature value, $\lambda \in [1.0, 2.0]$, that provides good balance. When exploration is needed, set $\lambda \leq 0.5$ for higher model diversity. On the other hand, for exploitation focus, set $\lambda \geq 5.0$ when confident in model ranking.

## C.2 EXPLORATION RATE $\epsilon_n$ DETAILED ANALYSIS

### C.2.1 EXPERIMENTAL SETUP

We conducted comprehensive ablation studies testing 30 configurations (6 epsilon values $\times$ 5 decay strategies) across 6 datasets with 10 trials each, totaling 1,800 experiments. The datasets span diverse prediction tasks: Sarcos Robot (trajectory prediction), Abalone (age prediction), Synthetic Mixture (heteroscedastic), Medical Cost (insurance), Bike Sharing (demand), and Parkinson's (tele-monitoring).

**Model Ensemble.** To maximize potential exploration benefits, we tested with 3 Random Forests (max_depth $\in \{3, 5, 10\}$), 2 Gradient Boosting (learning_rate $\in \{0.01, 0.1\}$), 3 Ridge Regression ($\alpha \in \{0.01, 1.0, 100.0\}$), 1 Decision Tree (max_depth = 8), and 1 K-Nearest Neighbors ($k = 20$).

**Decay Strategies.** We implemented five epsilon decay strategies:

$$\text{Constant:} \quad \epsilon_t = \epsilon_0 \tag{76}$$

$$\text{Square Root:} \quad \epsilon_t = \epsilon_0/\sqrt{t} \tag{77}$$

$$\text{Logarithmic:} \quad \epsilon_t = \epsilon_0 \sqrt{\log(K)/t} \tag{78}$$

$$\text{Linear:} \quad \epsilon_t = \epsilon_0 \left(1 - 0.99 \min(t/1000, 1)\right) \tag{79}$$

$$\text{Exponential:} \quad \epsilon_t = \epsilon_0 \exp(-0.01t/100) \tag{80}$$

### C.2.2 KEY FINDINGS

Table 16: Complete exploration rate results (mean ± std across 60 trials per configuration)

| $\epsilon$ | Coverage | Norm. Width | Regret | Models Used | Selection Entropy |
|---|---|---|---|---|---|
| 0.00 | 0.899 ± 0.016 | **2.193 ± 0.987** | **0.171 ± 0.052** | 4.3 | 1.538 |
| 0.01 | 0.899 ± 0.016 | 2.193 ± 0.987 | 0.172 ± 0.052 | 5.8 | 1.550 |
| 0.05 | 0.899 ± 0.016 | 2.200 ± 0.983 | 0.176 ± 0.051 | 7.9 | 1.594 |
| 0.10 | 0.899 ± 0.015 | 2.199 ± 0.975 | 0.180 ± 0.050 | 8.6 | 1.644 |
| 0.20 | 0.899 ± 0.016 | 2.216 ± 0.964 | 0.189 ± 0.048 | 9.2 | 1.733 |
| 0.50 | 0.900 ± 0.016 | 2.275 ± 0.907 | 0.219 ± 0.054 | 9.8 | 1.947 |

**Heatmap Analysis.** In Figure 6 (top left), the normalized width remains stable (2.193) across most configurations. Only 4 cells exceed the significance threshold of 2.243 (baseline $+0.05$): $\epsilon = 0.2$ with constant decay (2.281), and $\epsilon = 0.5$ with constant (2.305), exponential (2.256), and linear (2.446) decay. This concentration of degradation at high exploration rates confirms excessive exploration hurts performance.

**Model Diversity Without Benefit.** While unique models selected increases monotonically with $\epsilon$, the correlation with exploration rate is not statistically significant ($\rho = 0.75$, $p = 0.086$). Pure exploitation uses only 43% of available models yet achieves optimal performance, demonstrating that diversity alone does not improve conformal prediction (Figure 6, top right).

**Exploration-Exploitation Tradeoff.** As shown in Figure 4, both metrics degrade with exploration. Width increases 3.7% from $\epsilon = 0$ (2.193) to $\epsilon = 0.5$ (2.275). Regret increases 40% from $\epsilon = 0$ (0.171) to $\epsilon = 0.5$ (0.219). Coverage remains valid (89.9-90.0%) across all configurations.

**Temperature Interaction.** All three $\lambda$ configurations (adaptive, fixed = 1.0, fixed = 2.0) show similar trajectories, with differences $< 2\%$ at low exploration rates. This convergence suggests the temperature mechanism dominates selection dynamics regardless of exploration strategy (Figure 6, bottom left).

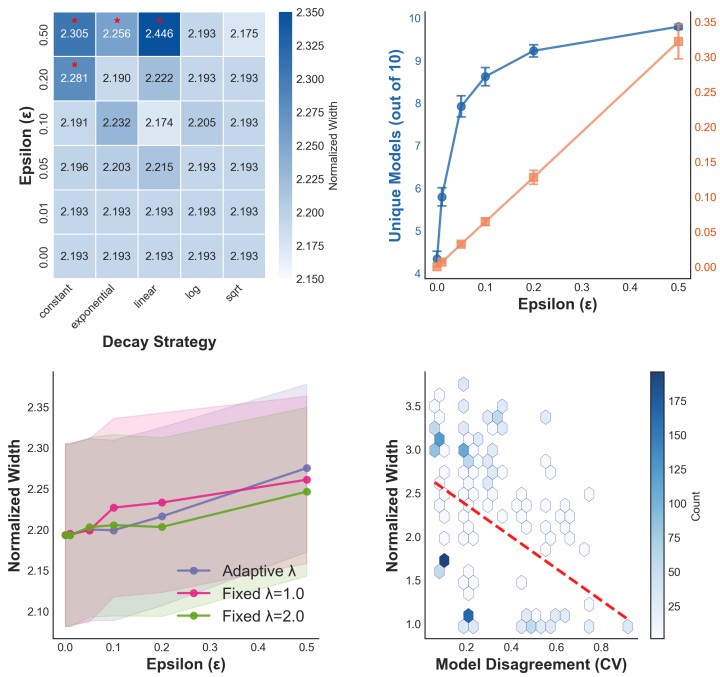

Figure 6: Exploration rate ablation results. From top left to bottom right, normalized width heatmap showing minimal variation ($2.193 - 2.446$) with stars marking significant degradation ($>2.243$). Model selection diversity increases from 4.3 to 9.8 models ($\rho = 0.75$, $p = 0.086$). Temperature interaction shows convergent patterns. Model disagreement negatively correlates with performance ($\rho = -0.364$, $p < 0.001$). Summary table 16 confirm pure exploitation optimality.

**Model Disagreement Paradox.** Counterintuitively, model disagreement negatively correlates with performance ($\rho = -0.364$, $p < 0.001$, $n = 1686$). Higher ensemble diversity leads to wider intervals, contradicting conventional wisdom that diverse models improve ensemble methods. The hexbin visualization reveals clustering at low disagreement ($CV < 0.4$) with optimal performance (Figure 6, bottom right).

**Dataset-Specific Patterns.** Only high-dimensional datasets (Sarcos Robot Arm) benefit from exploration, while most datasets perform optimally with pure exploitation (Table 17).

Table 17: Optimal exploration strategy by dataset characteristics

| Dataset | Best $\epsilon$ | Width | Model CV | Characteristics |
|---|---|---|---|---|
| Bike Sharing | 0.0 | 0.966 | 0.588 | High noise, temporal patterns |
| Parkinson's | 0.0 | 1.582 | 0.099 | Low model disagreement |
| Synthetic Mixture | 0.2 | 3.007 | 0.066 | Heteroscedastic noise |
| Sarcos Robot Arm | 0.5 | 2.268 | 0.351 | High-dimensional (21 features) |

**Theoretical Implications.** These results validate that the temperature-scaled posterior provides sufficient implicit exploration. The posterior distribution:

$$\rho_i \propto \pi_i \exp\left(-\lambda\, Q_{1-\alpha}[s_i]\right)$$

naturally balances exploitation of low-quantile models with uncertainty-driven exploration through the soft-max mechanism. Additional $\epsilon$-greedy exploration disrupts this balance without improving selection quality.

**Practical Recommendations.** Based on 1,800 experiments across diverse settings:

1. *Default:* Set $\epsilon = 0$ (pure exploitation) for optimal performance

2. *High-dimensional problems:* Consider $\epsilon \leq 0.1$ with sqrt decay only if $d > 20$
3. *Avoid:* $\epsilon > 0.2$ which consistently degrades performance (up to 11% width increase)
4. *Robustness:* All configurations maintain valid coverage, ensuring safety even with suboptimal choices

The surprising effectiveness of pure exploitation simplifies PBIS deployment, eliminating a hyper-parameter while maintaining theoretical guarantees and optimal empirical performance.

### C.3 PRIOR DISTRIBUTION ANALYSIS DETAILS

#### C.3.1 EXPERIMENTAL SETUP

We evaluated prior impact across six datasets: Sarcos Robot Arm (21-dimensional trajectory prediction), Abalone (age prediction), Synthetic Mixture (nonlinear regression), Medical Cost (insurance cost prediction), Bike Sharing (demand forecasting), and Concrete Strength (material science regression). These datasets span different domains and complexity levels.

#### C.3.2 PRIOR SPECIFICATIONS

We tested eight prior distributions over the model ensemble:

- **Uniform**: Equal weight across all models ($\pi_i = 1/K$)
- **Complexity**: Exponentially decreasing weight by model complexity, favoring Ridge/Lasso over RF/GB
- **Performance**: Weights based on validation set quantiles using 30% of training data
- **Maximum Entropy**: Bell-shaped distribution centered on medium-complexity models
- **Random**: Dirichlet-distributed random weights (sensitivity baseline)
- **Misspecified (3 levels)**: Deliberately incorrect priors favoring complex models

| Prior Type | Coverage | Width | Std(Width) | KL Divergence | Prior Entropy |
|---|---|---|---|---|---|
| Uniform | 0.917 | 1384 | 2941 | 0.42 | 2.303 |
| Complexity | 0.914 | 1758 | 4242 | 0.55 | 1.792 |
| Performance | 0.924 | 1480 | 3178 | 0.06 | 1.223 |
| Entropy | 0.909 | 1415 | 3050 | 0.42 | 1.530 |
| Random | 0.916 | 1575 | 3410 | 0.39 | 1.975 |
| Misspec-Mild | 0.921 | 1370 | 2919 | 0.47 | 2.250 |
| Misspec-Moderate | 0.925 | 1399 | 3016 | 0.58 | 1.944 |
| Misspec-Severe | 0.926 | 2728 | 6757 | 0.53 | 0.466 |

Table 18: Complete prior ablation results averaged across all calibration sizes and datasets. All configurations maintain valid coverage (target: 0.90). KL divergence measures information gain from prior to posterior.

**Model Selection Patterns.** Different priors induce distinct selection behaviors. The uniform prior induces balanced selection (GB-5: 34.3%, Lasso-0.5: 13.9%, RF-8: 12.6%). The complexity strongly favors simple models (Lasso-0.1: 41.3%, Ridge-0.1: 25.7%). The performance prior concentrates on best performers (GB-5: 45.2%, Lasso-0.5: 17.8%).

**Robustness Analysis.** As shown in Table 18, PBIS demonstrates remarkable robustness to prior misspecification. Even under severe misspecification (99% weight on worst model), coverage remains valid (92.6%) with only $2\times$ width increase. This robustness stems from the quantile-aware posterior construction (Equation 8), which allows data to override poor prior choices through the temperature-scaled likelihood term.

**Interpretation.** The limited impact of prior selection aligns with our theoretical analysis. As calibration size increases, the posterior converges to the empirical quantile-minimizing model at rate $O(n^{-1/2})$ (Section 5), making prior influence negligible for $n \geq 200$. The counterintuitive under-performance of complexity priors with limited data suggests that matching prior bias to problem structure requires domain knowledge that may not be available in practice.