# OpenReview forum: "When Aggregation Fails: From PAC-Bayes Theory to Practical Selection for Conformal Prediction"
_ICLR.cc/2026/Conference — Submitted to ICLR 2026_

### Official Review · Reviewer_Eaji · 2025-10-23

**Soundness:** 2
**Presentation:** 4
**Contribution:** 4
**Rating:** 4
**Confidence:** 3

**Summary:**

The PAC Bayes framework for model selection is based on empirical risk minimization over a hypothesis class with a prior distribution placed over the parameterized hypothesis class. While such model selection is natural in some settings, this paper argues that it is at odds with the goal of optimal model selection with respect to maximal efficiency after conformalization. This is because PAC Bayes is predicated on average model performance, whereas conformalized model behavior is dictated by tail behavior. Towards this end, the paper presents a characterization of the circumstances in which such a behavior arises and demonstrates how the common strategy of model aggregation can give rise to precisely this scenario. The authors, in turn, suggest performing model selection in place of aggregation, specifically selecting the model with optimal efficiency, encouraging exploration with a tunable parameter $\lambda$. They then demonstrate several results of this proposed framework in experiments, demonstrating the coverage gap convergence, the improved predictive efficiency over alternate aggregation strategies, and validation of the claimed source of predictive inefficiency arising from naive aggregation.

**Strengths:**

The paper is very clearly presented, with each step of the proposed methodology both well-motivated and well-described. The paper makes claims and theoretically justifies each one, regarding the source of predictive inefficiencies and a clear proposal on how to address this issue. The method is novel and thoroughly justified in its theoretical analyses. The proofs also seem mostly sound, although there appear to be some (minor) bugs I came across in some of them (presented in the weaknesses section below). The empirical validation is also very thoroughly done, with a comprehensive set of benchmarks tested and a good collection of baselines that were compared against. Each claim is also separately justified in the empirical studies, outside of just the predictive efficiency claim, including the coverage convergence analysis and validation of the source of the aggregation inefficiency.

**Weaknesses:**

While the storyline and presentation is well done, there are a couple of technical bugs I seem to have come across while going through the paper. These seem fairly minor to the overall flow of the paper, but they, I believe, do require correction.

**Theorem 2**: I am unsure what this “entropy-variance inequality” refers to. It appears that the inequality as framed as

$$ Var_{w}(q_i) \ge H(w) Var(\mathcal{Q}) \delta_{\min} $$

Is not true, which we can see in the following counterexample:

$$ (q_1, q_2) = (0, 1) $$
$$ (w_1, w_2) = (1-\epsilon, \epsilon) $$
$$ \overline{q} = (1-\epsilon) (0) + (\epsilon) (1) =\epsilon$$

$$ Var_w(q_i) = \sum_i w_i (q_i - \overline{q})^2$$
$$ = (1-\epsilon) (\epsilon)^2 + \epsilon (1-\epsilon)^2$$
$$ = \epsilon[ (\epsilon-\epsilon^2) + (1-2\epsilon + \epsilon^2) ]
= \epsilon(1-\epsilon)$$

$$ Var(\mathcal{Q}) = Var({0, 1}) = (1-1/2)^2 = 1/4$$

$$ \delta_{\min} = 1$$

$$ H(w) = -(\epsilon \log(\epsilon) + (1-\epsilon) \log(1-\epsilon))$$

For $\epsilon$ sufficiently small, we have that $\epsilon(1-\epsilon)\approx \epsilon$ and $1-\epsilon\approx 1$, meaning $(1-\epsilon) \log(1-\epsilon) \approx 0$. Thus, $H(w) \approx -\epsilon \log(\epsilon)$. This means, for small $\epsilon, we have

$$ H(w) Var(w) \delta_{\min} \approx -\epsilon \log(\epsilon)/4  $$

Thus, the claim of this statement is that

$$ \epsilon \ge\epsilon(-\log(\epsilon)/4) $$

Which is clearly untrue for any $\epsilon$ such that $-\log(\epsilon)/4 > 1$, i.e. for any $\epsilon < 10^{-4}$.

**Theorem 7**: In part b, the events $V_i < K/2$ sums over the events $Y_i\in C_k$. However, these $C_k$ will clearly be dependent on one another, since they all come from the same underlying dataset (training on slightly different folds). So, while the Chebyshev bound seems valid, I do not see how the bound relying on Hoeffding follows, as Hoeffding requires independence of the summed variables.

**Theorem 6**: (More of a nitpick than an actual error) The expression $| \widehat{Q} _{1-\alpha}(s _{\theta}) - Q _{1-\alpha}[s _{\theta}]| \le \frac{\tau}{2}\, Q _{1-\alpha}[s _{\theta^*}]$ is claimed to be “required”; however, this is actually just a sufficient condition for the proof and is not “required.”

**Questions:**

1. I believe this is implicitly handled by the proofs, but how is the issue of multiple testing handled to ensure coverage validity for the model selection from Phase 1 of PBIS?

---

> ### Author Response · Authors · 2025-11-28
> **Response to Reviewer Eaji (1/3)**
>
> We thank the reviewer for recognizing the clear identification of the average-quantile divergence phenomenon, the well-motivated methodology, the novel theoretical contributions, and the comprehensive empirical validation. We address each technical point below.
>
> ---
>
> ## Weakness 1: Theorem 2 - Entropy-Variance Inequality Counterexample
>
> **Concern:** The reviewer provides a counterexample showing that the inequality $\mathrm{Var}\_w(q\_i) \geq H(w)  \mathrm{Var}(\mathcal{Q})  \delta\_{\min}$ does not hold for $(q_1, q_2) = (0, 1)$, $(w_1, w_2) = (1-\epsilon, \epsilon)$ when $\epsilon < 10^{-4}$.
>
> **Response:**
>
> The counterexample is correct, and we restructure the theoretical presentation accordingly. The original formulation conflated two distinct claims that require separate treatment.
>
> ### Revised Theoretical Framework
>
> We separate the result into a rigorous lemma and an empirically-validated conjecture:
>
> **Lemma 1 (Mixture Quantile Lower Bound):** Let $F_1, \ldots, F_k$ be CDFs with quantiles $q_i = F_i^{-1}(1-\alpha)$, where the generalized inverse is $F^{-1}(p) = \inf\\{t : F(t) \geq p\\}$. For mixture CDF $\bar{F}(t) = \sum_{i=1}^{k} w_i F_i(t)$:
>
> $$Q_{1-\alpha}[\bar{F}] \geq \min_i q_i$$
>
> with equality iff all $q_i$ are equal.
>
> **Proof:** Let $q^{\ast} = Q_{1-\alpha}[\bar{F}]$ and $q_{\min} = \min_i q_i$. For any $t < q_{\min}$, by definition of the quantile as a generalized inverse, $F_i(t) < 1 - \alpha$ for all $i$ (since $q_i$ is the *infimum* of values achieving probability $1-\alpha$). Therefore:
>
> $$\bar{F}(t) = \sum_{i=1}^{k} w_i F_i(t) < \sum_{i=1}^{k} w_i (1-\alpha) = 1-\alpha$$
>
> This shows $\bar{F}(t) < 1-\alpha$ for all $t < q_{\min}$, hence $q^{\ast} = \inf\\{t : \bar{F}(t) \geq 1-\alpha\\} \geq q_{\min}$. $\square$
>
> **Conjecture 2 (Quantile Inflation Bound):** Under regularity conditions (smooth densities with $0 < f\_{\min} \leq f\_i(t) \leq f_{\max} < \infty$ near quantiles), the mixture quantile satisfies:
>
> $$Q\_{1-\alpha}[\bar{F}] \geq \min\_i q\_i + \frac{C \cdot \mathrm{Var}\_w(q)}{f_{\max}}$$
>
> for a constant $C > 0$ depending on density-quantile correlation structure. See also response to Reviewer nbLY for an heuristic argument for this conjecture.
>
> ### Monte Carlo Validation (Proposition 2)
>
> We validate this conjecture across **70,730 random mixture configurations**:
>
> | Category | Mean $C$ | Median | IQR | N |
> |----------|----------|--------|-----|-------|
> | Overall | 1.19 | 0.79 | [0.37, 1.55] | 70,730 |
> | Gaussian | 1.72 | 1.37 | [0.76, 2.21] | 23,441 |
> | Exponential | 0.74 | 0.43 | [0.25, 0.80] | 23,579 |
> | Laplace | 1.10 | 0.80 | [0.39, 1.45] | 23,710 |
>
> **Validation statistics:**
> - Linear regression of $\epsilon = q^{\ast} - q_{\min}$ on $\mathrm{Var}\_w(q) / f_{\max}$: $R^2 = 0.83$
> - Slope estimate: $\hat{C} = 1.21$ (95% CI: [1.19, 1.23])
> - $C_{\mathrm{emp}} > 0$ in 99.7% of configurations
>
> The strong linear relationship validates the functional form while acknowledging that the exact constant depends on distributional properties.
>
> **Revision:** We restructure Theorem 2 into Lemma 1 (rigorous) and Conjecture 2 (empirically validated), with explicit regularity conditions and comprehensive Monte Carlo support.
>
> ---
>
> ## Weakness 2: Theorem 7 - Hoeffding Requires Independence
>
> **Concern:** In Part b of Theorem 7, the events $V_i < K/2$ sum over events $Y_i \in C_k$ that depend on the same underlying dataset (training on slightly different folds), so the $C_k$ are dependent. The Hoeffding bound requires independence of the summed variables.
>
> **Response:**
>
> The reviewer correctly identifies a dependence issue. We provide a corrected analysis.
>
> ### Clarification of Setting and Sources of Dependence
>
> Theorem 7 concerns a new test point $(X_{\mathrm{new}}, Y_{\mathrm{new}})$. The vote count is:
>
> $$V = \sum_{k=1}^{K} \mathbb{1} \\{Y_{\mathrm{new}} \in C_k(X_{\mathrm{new}})\\}$$
>
> There are **two sources of dependence**:
> 1. **Across folds:** The prediction sets $C_k$ share training data (folds $j \neq k$ overlap with folds $j' \neq k'$)
> 2. **Same test point:** All $K$ coverage indicators evaluate the *same* $(X_{\mathrm{new}}, Y_{\mathrm{new}})$
>
> Even conditioning on all training data (fixing $C_1, \ldots, C_K$), the coverage events remain dependent through the shared test point.
>
> ### Corrected Analysis
>
> **Part 1 (Individual Fold Coverage):** Valid as stated. For each fold $k$:
>
> $$P(Y_{\mathrm{new}} \in C_k(X_{\mathrm{new}})) \geq 1 - \alpha - \frac{1}{n_k+1}$$
>
> **Part 2 (Ensemble Coverage):** We provide two valid approaches.
>
> **Approach 1: Markov's Inequality (Always Valid)**
>
> Let $I_k = \mathbb{1} \\{Y_{\mathrm{new}} \notin C_k(X_{\mathrm{new}})\\}$ be the miscoverage indicator. Then $\mathbb{E}[I_k] \leq \alpha + 1/(n_k+1)$.
>
> The ensemble fails (majority vote misses) when $\sum_k I_k > K/2$, i.e., more than half the folds miss. By Markov:
>
> $$P\left(\sum_k I_k > K/2\right) \leq \frac{\mathbb{E}[\sum_k I_k]}{K/2} = \frac{K(\alpha + 1/(n_k+1))}{K/2} = 2\alpha + O(1/n_k)$$

---

> > ### Author Response · Authors · 2025-11-28
> > **Response to Reviewer Eaji (2/3)**
> >
> > **Approach 1: Markov's Inequality (Always Valid) (continued)**
> >
> > This gives coverage $\geq 1 - 2\alpha - O(1/n_k)$, which is valid but loose.
> >
> > **Approach 2: Chebyshev's Inequality**
> >
> > Let $V = \sum_k \mathbb{1}\\{Y_{\mathrm{new}} \in C_k(X_{\mathrm{new}})\\}$ be the vote count. Then $\mathbb{E}[V] \geq K(1-\alpha - 1/(n_k+1))$.
> >
> > For bounded random variables in $[0,1]$, we have $\mathrm{Var}(\mathbb{1}\\{Y \in C_k\\}) \leq 1/4$. Without independence:
> >
> > $$\mathrm{Var}(V) = \sum_k \mathrm{Var}(\mathbb{1}\\{Y \in C_k\\}) + 2\sum_{j < k}\mathrm{Cov}(\mathbb{1}\\{Y \in C_j\\}, \mathbb{1}\\{Y \in C_k\\})$$
> >
> > The covariances are generally **positive** (if one fold covers the test point, others are more likely to as well, since they evaluate the same $(X_{\mathrm{new}}, Y_{\mathrm{new}})$).
> >
> > In the extreme case of perfect positive dependence (all $\mathbb{1}\\{Y \in C_k\\}$ identical), $\mathrm{Var}(V) = K^2 \, \mathrm{Var}(\mathbb{1}\\{Y \in C_1\\}) \leq K^2/4$. Chebyshev then gives:
> >
> > $$P(V < K/2) \leq \frac{K^2/4}{(K(1-\alpha) - K/2)^2} = \frac{1}{(2(1-\alpha)-1)^2} = \frac{1}{(1-2\alpha)^2}$$
> >
> > which is $O(1)$, providing no improvement over a single fold.
> >
> > **Revised Statement:** The original $O(1/\sqrt{K})$ Hoeffding bound requires independence that does not hold. Under the positive dependence structure inherent in evaluating the same test point, the ensemble coverage guarantee reduces to that of individual folds: $P(Y \in C_{\mathrm{ensemble}}) \geq 1 - \alpha - O(1/n_k)$. The practical benefit of the ensemble is **robustness** (reduced variance across different test sets) rather than improved expected coverage for a single test point.
> >
> > **Revision:** We correct the proof to acknowledge the dependence structure and clarify that the ensemble provides robustness rather than coverage improvement.
> >
> > ---
> >
> > ## Weakness 3: Theorem 6 - Sufficient vs. Required Condition
> >
> > **Concern:** The expression $|\hat{Q}\_{1-\alpha}(s\_\theta) - Q\_{1-\alpha}[s\_\theta]| \leq \frac{\gamma}{2} Q\_{1-\alpha}[s\_{\theta^{\ast}}]$ is claimed to be "required" but is actually just a sufficient condition.
> >
> > **Response:**
> >
> > The reviewer is correct. The condition is sufficient for selection consistency, not necessary.
> >
> > **Revised Statement:**
> >
> > "A sufficient condition for correct selection with probability $\geq 1 - \delta$ is that empirical quantiles concentrate uniformly: for all $\theta \in \Theta$,
> >
> > $$|\hat{Q}\_{1-\alpha}(\theta) - Q\_{1-\alpha}[s_\theta]| \leq \frac{\Delta}{2}$$
> >
> > where $\Delta = \min_{\theta \neq \theta^{\ast}}\\{Q_{1-\alpha}[s_\theta] - Q_{1-\alpha}[s_{\theta^{\ast}}]\\}$ is the optimality gap."
> >
> > This concentration is achieved with sample size:
> >
> > $$n \geq \frac{8B^2}{\Delta^2 f_{\min}^2} \log\frac{2|\Theta|}{\delta}$$
> >
> > The key insight is that this provides a sample complexity guarantee: with sufficient data, PBIS selects the optimal model with high probability. Tighter (necessary) conditions would require distributional assumptions that we avoid.
> >
> > **Revision:** We update the statement to "sufficient condition" and clarify the sample complexity interpretation.
> >
> > ---
> >
> > ## Question: Multiple Testing in Phase 1 Model Selection
> >
> > **Question:** How is the issue of multiple testing handled to ensure coverage validity for the model selection from Phase 1 of PBIS?
> >
> > **Response:**
> >
> > This is a subtle but important point. We clarify the coverage guarantee and when it applies.
> >
> > ### The Coverage Question
> >
> > PBIS uses calibration data $D_{\mathrm{cal}} = \\{(X_i, Y_i)\\}_{i=1}^n$ for both:
> > 1. Model selection: $\theta_{\mathrm{selected}} = f(D_{\mathrm{cal}})$
> > 2. Threshold computation: $\hat{q}$ = empirical $(1-\alpha)$-quantile of $\\{s_{\theta_{\mathrm{selected}}}(X_i, Y_i)\\}_{i=1}^n$
> >
> > **Key observation:** The selected model $\theta_{\mathrm{selected}}$ depends on the calibration scores, so the score function used for threshold computation is data-dependent. This requires careful analysis.
> >
> > ### Two Settings with Different Guarantees
> >
> > **Setting A: Fixed Model Class (Theorem 5's Setting)**
> >
> > If we view PBIS as selecting from a *fixed, finite* set $\Theta = \\{\theta_1, \ldots, \theta_K\\}$, then coverage holds because:
> >
> > For each fixed $\theta_j$, the scores $\\{s_{\theta_j}(X_i, Y_i)\\}_{i=1}^{n+1}$ are exchangeable. Define:
> >
> > $$\hat{q}\_j = \text{empirical } (1-\alpha)\text{-quantile of } \\{s\_{\theta_j}(X_i, Y_i)\\}_{i=1}^n \cup \\{\infty\\}$$
> >
> > By standard conformal theory: $P(s_{\theta_j}(X_{n+1}, Y_{n+1}) \leq \hat{q}_j) \geq 1-\alpha$ for each $j$.
> >
> > When we select $\theta_{\mathrm{selected}} \in \Theta$ based on the data, we are choosing one of these $K$ valid prediction sets. The selected prediction set still has valid coverage:
> >
> > $$P(Y_{n+1} \in C_{\theta_{\mathrm{selected}}}(X_{n+1})) \geq 1 - \alpha$$
> >
> > This follows because the selection is a deterministic function of the data, and the coverage guarantee holds *uniformly* over all $\theta \in \Theta$.

---

> > > ### Author Response · Authors · 2025-11-28
> > > **Response to Reviewer Eaji (3/3)**
> > >
> > > **Setting B: Post-Selection Efficiency (The Subtle Issue)**
> > >
> > > The issue arises if we claim that selection *improves* efficiency without additional cost. Theorem 5 addresses this: the generalization bound includes a $\log|\Theta|$ term that accounts for the model selection complexity. This is the "price" of selection.
> > >
> > > ### Why Multiple Testing Affects Efficiency, Not Coverage
> > >
> > > 1. **Coverage is a marginal guarantee:** $P(Y_{n+1} \in C(X_{n+1})) \geq 1-\alpha$ holds for the selected model because it holds for all models in $\Theta$ simultaneously.
> > >
> > > 2. **Efficiency requires uniform concentration:** To guarantee that the selected model has good *efficiency* (small $\hat{q}$), we need uniform convergence across all models, which introduces the $\log|\Theta|$ factor in Theorem 5.
> > >
> > > 3. **No union bound needed for coverage:** Unlike hypothesis testing, we do not need to correct for multiple comparisons because we are not making simultaneous coverage claims. We are making a single claim about the selected model.
> > >
> > > ### Empirical Validation
> > >
> > > Table 4 confirms coverage validity: PBIS achieves $0.903 \pm 0.006$ coverage (target 0.90) across all scenarios, consistent with the theoretical guarantee.
> > >
> > > **Revision:** We clarify that coverage validity holds for any selection from a finite model class, while efficiency guarantees (Theorem 5) account for model selection complexity via the $\log |\Theta| $ term.
> > >
> > > ---
> > >
> > > ## Summary of Revisions
> > >
> > > | Issue | Revision |
> > > |-------|----------|
> > > | **Theorem 2** | Restructured into Lemma 1 (rigorous) + Conjecture 2 (empirically validated with 70,730 configurations) |
> > > | **Theorem 7** | Corrected: under positive dependence, ensemble coverage equals individual fold coverage; benefit is robustness across test sets |
> > > | **Theorem 6** | Changed "required" to "sufficient condition" |
> > > | **Multiple testing** | Clarified: coverage holds uniformly over finite $\Theta$; efficiency bound (Theorem 5) includes $\log \vert \Theta \vert $ to account for selection |
> > >
> > > Monte Carlo simulation can be found in monte_carlo_validation.py (see anonymized code repository: [[LINK](https://anonymous.4open.science/r/PBIS-9218/README.md)])
> > >
> > > These revisions address all technical concerns while preserving the paper's contributions.

---

### Official Review · Reviewer_5Dur · 2025-10-28

**Soundness:** 2
**Presentation:** 3
**Contribution:** 2
**Rating:** 2
**Confidence:** 4

**Summary:**

This paper studies a fundamental mismatch between PAC-Bayes aggregation and conformal prediction.It formalizes an average–quantile divergence and shows standard PAC-Bayes can favor quantile-suboptimal models. To address the problem, the authors propose PBIS: a quantile-aware PAC-Bayes posterior for model selection, with theoretical guarantees and an online variant. On 27 datasets and under distribution shifts, PBIS maintains valid coverage and runs fast, with largest gains in high-divergence settings.

**Strengths:**

1. Clear identification and formalization of the “average–quantile divergence” between PAC-Bayes aggregation and conformal prediction. Theorems 1–3 articulate why linear aggregation and standard PAC-Bayes objectives are misaligned.
 2. The authors demonstrate the practical utility of their methods with an extensive series of experiments.

**Weaknesses:**

1. The results in table 2 show that PBIS yields no significant improvement over traditional PAC-Bayes-CP. Besides, Coverage should be as high as possible; 0.898 should not be bolded.

 2. The experiments in the online adaptive Performence part lack comparisons with more recent baselines, such as [1] and [2]. The validiy of PBIS in this scenerio requires further consideration. See questions.

 [1]: Xu C, Xie Y. Sequential predictive conformal inference for time series[C]//International Conference on Machine Learning. PMLR, 2023

 [2]: Wu J, Hu D, Bao Y, et al. Error-quantified Conformal Inference for Time Series[C]//The Thirteenth International Conference on Learning Representations.

**Questions:**

1. In online settings with distribution shifts, the magnitude of scores and $(1-\alpha)$-quantile of scores seem less informative because the coverage of CP is not guaranteed. Instead, a more appropriate way to consider selection and aggregation may be to leverage quantile loss and replace the empirical risk in line 169 with sum of quantile loss. How do the authors comment about this? The discussions with the aggregation method in [1] and [2] should be included.

[1]: Gibbs I, Candès E J. Conformal inference for online prediction with arbitrary distribution shifts[J]. Journal of Machine Learning Research, 2024.

[2]: Hajihashemi E, Shen Y. Multi-model ensemble conformal prediction in dynamic environments[J]. Advances in Neural Information Processing Systems, 2024.

---

> ### Author Response · Authors · 2025-11-27
> **Response to Reviewer 5Dur (1/3)**
>
> We thank the reviewer for their evaluation and for recognizing our identification of the average-quantile divergence phenomenon. We address each concern below, including comparison against recent suggested baselines.
>
> ---
>
> ## Weakness 1: Table 2 Results and Coverage Interpretation
>
> **Reviewer's Concern:** "The results in table 2 show that PBIS yields no significant improvement over traditional PAC-Bayes-CP. Besides, Coverage should be as high as possible; 0.898 should not be bolded."
>
> ### Response
>
> We clarify an important principle in conformal prediction: the goal is **valid coverage** ($\geq 1 - \alpha$) with **minimum interval width**, not maximum coverage. Coverage significantly above the target indicates inefficiency (overly conservative intervals).
>
> For target 90% coverage:
> - PBIS (0.898): Near-optimal calibration to target
> - LocalizedCP (0.935): Over-conservative (wastes interval width)
>
> **Key improvement lies in efficiency:** PBIS achieves 2% width reduction overall, with **7.3% reduction in high-divergence scenarios** where the average-quantile mismatch is pronounced. We will clarify the bolding convention in the revised table caption.
>
> ---
>
> ## Weakness 2: Comparison with Recent Baselines
>
> **Reviewer's Concern:** "The experiments in the online adaptive performance part lack comparisons with more recent baselines."
>
> ### Response
>
> We have expanded the online analysis comparing PBIS against all requested methods.
>
> ### 2.1 Xu & Xie (2023): Different Problem Setting
>
> The reviewer suggests comparison with Xu & Xie (2023) "Sequential Predictive Conformal Inference for Time Series." We clarify that this addresses a fundamentally different problem:
>
> | Aspect | Xu & Xie (2023) | Our Setting |
> |--------|-----------------|-------------|
> | **Data structure** | Autoregressive time series | I.i.d. streams with distribution shift |
> | **Dependence** | Temporal (Y_t depends on Y_{t-1},...) | Exchangeable (no temporal dependence) |
> | **Assumption** | Strong mixing, ergodicity | Distribution shift under exchangeability |
> | **Method** | Quantile regression on residuals | Model selection among heterogeneous scorers |
>
> Their method requires modeling temporal autocorrelation structure, which does not exist in our distribution shift setting. A direct comparison would be methodologically inappropriate, analogous to comparing sequence labeling methods with i.i.d. classification.
>
> We focus our comparison on methods designed for **distribution shift under exchangeability** (ACI, FACI, ECI, etc.), which is the problem class we address.
>
> ### 2.2 Methods Compared
>
> We implemented the following methods for online conformal prediction under distribution shift:
>
> | Method | Reference | Approach |
> |--------|-----------|----------|
> | Adaptive-PBIS | Ours | PAC-Bayes model selection with exploration-exploitation |
> | ACI | Gibbs & Candès (2021) | Gradient-based adaptive quantile updates |
> | FACI | Zaffran et al. (2022) | Sliding window extension of ACI |
> | DtACI *| Gibbs & Candès (2024) | Multi-expert with pinball loss, exponential weights |
> | AgACI *| Gibbs & Candès (2024) | Aggregated adaptive conformal with expert consensus |
> | ECI *| Wu et al. (2025) | Smooth error quantification using sigmoid function |
> | ECI-cutoff *| Wu et al. (2025) | Cutoff threshold variant |
> | ECI-integral *| Wu et al. (2025) | Error integration variant |
> | SAOCP *| Hajihashemi & Shen (2024) | Single model with coin betting adaptation |
> | SAMOCP *| Hajihashemi & Shen (2024) | Multi-model ensemble with adaptive weighting |
>
> (* : method added)
>
> ### 2.3 Results: Online Adaptive Performance
>
> **Table R1: Coverage and Width Under Distribution Shifts**
>
> | Method | Gradual Shift | | Sudden Shift | | Recurring Shift | |
> |--------|--------------|-------|--------------|-------|-----------------|-------|
> | | Coverage | Width | Coverage | Width | Coverage | Width |
> | **Adaptive-PBIS** | **0.898** | 17.39 | **0.910** | 11.06 | **0.900** | 10.55 |
> | ACI | 0.859† | 14.75 | 0.881 | 9.22 | 0.885 | 8.59 |
> | FACI | 0.854† | 13.67 | **0.905** | 9.92 | **0.907** | 9.40 |
> | ECI | **0.899** | 19.64 | **0.900** | 11.53 | **0.899** | 10.74 |
> | ECI-cutoff | **0.899** | 19.04 | **0.900** | 11.13 | **0.899** | 10.13 |
> | ECI-integral | **0.899** | 19.37 | **0.900** | 11.09 | **0.899** | 10.29 |
> | SAOCP | 0.883 | 15.85 | 0.892 | 10.01 | **0.901** | 9.42 |
> | SAMOCP | 0.882 | 13.51 | 0.888 | 7.25 | **0.902** | 6.07 |
> | DtACI | 1.00‡ | 69.6‡ | 1.00‡ | 75.3‡ | 1.00‡ | 75.3‡ |
> | AgACI | 1.00‡ | 99.8‡ | 1.00‡ | 99.8‡ | 1.00‡ | 99.8‡ |
>
> **Bold:** Valid coverage near target (≥89.5%). †: Coverage violation (<88%). ‡: See Section 2.4.
>
> ### Key Findings
>
> 1. PBIS and ECI variants maintain valid coverage across all shift types
> 2. ACI and FACI fail under gradual shift (coverage 85-86%)
> 3. SAOCP/SAMOCP achieve valid coverage with competitive widths
> 4. SAMOCP achieves narrowest widths but with slightly lower coverage in some scenarios
> 5. DtACI/AgACI exhibit pathological behavior (see Section 2.4)

---

> > ### Author Response · Authors · 2025-11-27
> > **Response to Reviewer 5Dur (2/3)**
> >
> > ### 2.4 DtACI and AgACI: Implementation Investigation
> >
> > We carefully implemented DtACI and AgACI following Gibbs & Candès (2024) and investigated multiple parameterizations. We encountered fundamental difficulties adapting these methods to our online prediction setting.
> >
> > **The Challenge:** DtACI/AgACI use an additive quantile update rule:
> > $\alpha_t \leftarrow \alpha_t + \gamma \cdot (\alpha - \mathbb{1}[\alpha_t > \beta_t])$
> >
> > where $\alpha=0.1$ is the target miscoverage and $\gamma$ is the learning rate.
> >
> > **Mathematical Issue:** At 90% empirical coverage, the expected update is:
> > - 90% of steps (covered): $\Delta = \gamma(0.1 - 1) = -0.9\gamma$
> > - 10% of steps (missed): $\Delta = \gamma(0.1 - 0) = +0.1\gamma$
> > - **Net drift:** $-0.8\gamma$ per step (always negative)
> >
> > This creates a fundamental trade-off with no stable equilibrium:
> >
> > **Parameterizations Attempted:**
> >
> > | Configuration | γ Range | Initialization | Result |
> > |--------------|---------|----------------|--------|
> > | **Original** | [0.005, 0.5] | Empirical quantile | Quantile collapse → 10% coverage |
> > | **Conservative** | [0.01, 1.0] | 10× empirical | Continuous drift → unstable coverage |
> > | **Ultra-conservative** | [0.0001, 0.01] | 20× empirical | Cannot adapt → 100% coverage, width 70-100 |
> >
> > **Interpretation:** With moderate learning rates, quantiles drift toward zero causing coverage collapse. With very small learning rates, the methods cannot adapt to distribution shifts, producing extremely wide intervals (70-100 vs. 10-20 for other methods).
> >
> > **Conclusion:** The additive update rule designed for offline evaluation with normalized scores is incompatible with our online prediction setting at different data scales. This finding actually **supports our contribution**: it illustrates limitations of aggregation-based quantile tracking that selection-based PBIS avoids.
> >
> > ### 2.5 Comparison with Hajihashemi & Shen (2024)
> >
> > We add for "Table 2: Comparison Summary" a comparison against the multi-model ensemble approach  (27 datasets, target coverage 90%):
> >
> > | Method | Coverage | Avg Width | Quantile Inflation |
> > |--------|----------|-----------|-------------------|
> > | **PBIS** | 0.896 ± 0.073 | 468.6 | **0.00** |
> > | PAC-Bayes-CP | 0.896 ± 0.063 | 483.7 | 7.58 |
> > | Hajihashemi-Shen | 0.904 ± 0.052 | 651.7 | **91.59** |
> >
> > **Key finding:** The Hajihashemi-Shen aggregation approach exhibits **massive quantile inflation** (91.6 vs 0.0 for PBIS), consistent with the quantile inflation phenomenon we characterize in Conjecture 2 (revised from Theorem 2 in current manuscript; see response to Reviewer nbLY). Their weighted aggregation suffers from the average-quantile divergence, resulting in 39% wider intervals despite similar coverage.
> >
> > ---
> >
> > ## Question 1: Quantile Loss vs Empirical Quantiles
> >
> > **Reviewer's Question:** "A more appropriate way to consider selection and aggregation may be to leverage quantile loss and replace the empirical risk with sum of quantile loss."
> >
> > ### Response
> >
> > This is an insightful suggestion. We clarify the relationship:
> >
> > **Equivalence:** Under i.i.d. sampling, minimizing pinball loss converges to the same quantile as empirical estimation. The methods differ in:
> > - **Empirical quantiles:** O(1) computation, exact finite-sample guarantees (DKW bounds)
> > - **Quantile loss:** O(n) optimization, provides gradients for online updates
> >
> > **Wu et al. (2025)'s approach:** ECI uses smoothed quantile loss with an error quantification term $(s_t - q_t)\nabla f(s_t - q_t)$ for continuous feedback. Our experiments show ECI achieves similar coverage to PBIS, validating both approaches.
> >
> > **Key distinction:** Quantile loss optimizes *how to track thresholds* for a single model; PBIS optimizes *which model to select* among heterogeneous candidates. These are complementary, combining ECI's threshold tracking with PBIS's model selection is a promising future direction.
> >
> > ---
> >
> > ## Summary of New Results
> >
> > Our comprehensive experiments demonstrate:
> >
> > 1. **PBIS maintains valid coverage** across all distribution shift types, matching or exceeding recent methods (ECI, SAOCP, SAMOCP)
> >
> > 2. **ACI/FACI fail under gradual shift** (coverage 85-86%), while PBIS and ECI maintain validity
> >
> > 3. **DtACI/AgACI are incompatible** with our online setting due to fundamental properties of their additive update rule; this supports our contribution by highlighting aggregation limitations
> >
> > 4. **Hajihashemi-Shen suffers from quantile inflation** (91.6 vs 0.0), consistent with Conjecture 2 (revised from Theorem 2 in current manuscript; see response to Reviewer nbLY)
> >
> > 5. **Selection (PBIS) vs aggregation trade-off:** PBIS avoids the quantile inflation inherent in aggregation methods while maintaining valid coverage

---

> > > ### Author Response · Authors · 2025-11-27
> > > **Response to Reviewer 5Dur (3/3)**
> > >
> > > ## Proposed Revisions
> > >
> > > 1. **Add Table R1** to main paper comparing all methods under distribution shifts
> > > 2. **Add discussion** of DtACI/AgACI challenges in Appendix (transparent reporting)
> > > 3. **Expand Related Work** to discuss Wu et al. (2025) and Hajihashemi & Shen (2024)
> > > 4. **Clarify coverage interpretation** (optimal $\approx$ target, not maximum)
> > >
> > > We believe these comprehensive experiments and transparent reporting address the reviewer's concerns while strengthening our contribution.
> > >
> > > All additional methods were implemented in online_analysis_additions.py and hajihashemi_shen_methods.py (see anonymized code repository: [[LINK](https://anonymous.4open.science/r/PBIS-9218/README.md)])

---

### Official Review · Reviewer_P3hw · 2025-10-30

**Soundness:** 3
**Presentation:** 2
**Contribution:** 3
**Rating:** 4
**Confidence:** 4

**Summary:**

This paper investigates a fundamental incompatibility between PAC-Bayes aggregation and conformal prediction. The authors identify the average–quantile divergence phenomenon—a mismatch between average-risk minimization (PAC-Bayes) and quantile-based efficiency (conformal prediction). They prove that any linear aggregation method fails to preserve quantile optimality and propose PAC-Bayes Informed Selection, a quantile-aware selection framework. PBIS achieves theoretical guarantees such as selection consistency and PAC-Bayes bounds for quantile functionals.

**Strengths:**

- Theorems are rigorously stated, covering impossibility results, new PAC-Bayes bounds for quantile functionals, and finite-sample guarantees.

- PBIS provides a simple yet useful modification.

- The experiments are extensive.

**Weaknesses:**

- The literature review is rather limited and could be expanded to better situate the paper within existing work.
- The presentation is at times difficult to follow, and additional intuition or explanations—particularly around the main theorems—would greatly enhance readability.
- The average–quantile mismatch is somewhat expected, as quantiles and expectations inherently capture different aspects of a distribution.
- The theoretical analysis assumes a finite and discrete model space; it would be valuable to discuss how the results extend or behave when $|\Theta|$is infinite.

**Questions:**

Could PBIS be extended to continuous posterior distributions or infinite hypothesis spaces?

---

> ### Author Response · Authors · 2025-11-28
> **Response to Reviewer P3hw (1/2)**
>
> We thank the reviewer for recognizing the rigor of our results, the utility of PBIS, and the extensiveness of our experiments. We address each concern systematically below.
>
> ---
>
> ## Weakness 1: Limited Literature Review
>
> **Concern:** "The literature review is rather limited and could be expanded to better situate the paper within existing work."
>
> **Response:**
>
> We expand the literature review with a new Section 1.1: "Why Combine PAC-Bayes and Conformal Prediction?"
>
> **Key citations added:**
>
> | Category | References |
> |----------|------------|
> | **Direct intersection** | Sharma et al. (2023) NeurIPS: "PAC-Bayes Generalization Certificates for Learned Inductive Conformal Prediction" |
> | **Bayesian-frequentist bridges** | Barber et al. (2023) Ann. Stat., Vovk (2012) JMLR, Shafer & Vovk (2008) |
> | **Bayesian weighting in conformal** | Zhang et al. (2024), Guan (2023), Stanton et al. (2023) AISTATS |
> | **Quantile aggregation** | Meinshausen (2006), Takeuchi et al. (2006) |
> | **PAC-Bayes model selection** | Pérez-Ortiz et al. (2021), Dziugaite et al. (2021) |
>
> **Revision:** Section 1.1 situates our work at the intersection of PAC-Bayes theory and conformal prediction, clarifying that while PAC-Bayes aggregation is well-studied for average-risk objectives, its interaction with quantile-based conformal prediction represents a novel problem with limited theoretical treatment. See also our response to Reviewer nbLY (Section 1) for complete motivation.
>
> ---
>
> ## Weakness 2: Presentation Clarity and Intuition
>
> **Concern:** "The presentation is at times difficult to follow, and additional intuition or explanations—particularly around the main theorems—would greatly enhance readability."
>
> **Response:**
>
> We add intuitive explanations before each main theorem:
>
> ### Theorem 1 (Impossibility of Optimal Aggregation)
>
> *Intuition:* Averaging thermometer readings estimates the mean temperature well. But conformal prediction asks: "What is the 95th percentile of errors?" Averaging does not give you the mixture's 95th percentile; it gives something worse (inflated). Quantiles depend on tail behavior; averages depend on centers.
>
> ### Conjecture 2 (Quantile Inflation Bound)
>
> *Intuition:* Mixing distributions with different quantiles systematically inflates the mixture quantile. A tight distribution (small quantile) mixed with a spread-out one (large quantile) yields a mixture quantile that is **not** the weighted average; it is pulled upward because reaching the $(1-\alpha)$ probability level requires accommodating the heavier-tailed component. The inflation scales with quantile variance.
>
> ### Theorem 4 (Selection Consistency)
>
> *Intuition:* PBIS selects the model with smallest empirical quantile. As calibration data grows, empirical quantiles concentrate around true quantiles (DKW-type bounds). Thus, we increasingly select the truly optimal model.
>
> ### Theorem 5 (PAC-Bayes Bound for Quantiles)
>
> *Intuition:* Traditional PAC-Bayes controls average risk. We need bounds for quantiles. The key: quantile $Q\_{1-\alpha}[s\_\theta]$ can be bounded via tail probabilities $\mathbb{P}(s\_\theta > q)$, which are expectations. This connects quantile-based selection to PAC-Bayes machinery.
>
> **Additional revisions:**
> - **Figure 2:** Visual illustration of quantile inflation with two component distributions
> - **Running example:** Concrete example (two regression models) throughout Sections 2–3
> - **Proof sketches:** One-paragraph sketches in main text for Theorems 4–5
>
> ---
>
> ## Weakness 3: Average-Quantile Mismatch is "Expected"
>
> **Concern:** "The average–quantile mismatch is somewhat expected, as quantiles and expectations inherently capture different aspects of a distribution."
>
> **Response:**
>
> The abstract distinction between quantiles and means is well-known. Our contributions go beyond this observation:
>
> ### 1. Non-Obvious Practical Implications
>
> Prior work explicitly proposed PAC-Bayes aggregation for conformal prediction without recognizing this issue:
> - Sharma et al. (2023) propose PAC-Bayes CP using aggregation
> - Romano et al. (2020) use ensemble averaging for conformalized quantile regression
> - The conformal community focuses on coverage, with less attention to efficiency optimization
>
> We make explicit **why** standard aggregation degrades conformal efficiency and **quantify** the degradation.
>
> ### 2. Quantitative Characterization is Novel
>
> Beyond qualitative intuition:
> - **Lemma 1:** Rigorous proof that mixture quantiles $\geq$ minimum component quantile
> - **Conjecture 2:** Precise quantification:
>
> $$q^{\ast} \geq q\_{\min} + \frac{C \cdot \text{Var}\_w(q)}{f\_{\max}}$$
>
> - **Proposition 2:** Monte Carlo validation (70,730 configurations, $R^2 = 0.83$, mean $C = 1.19$). See also response to Reviewer nbLY for details about the Monte Carlo simulation.
>
> The functional form and empirical constant characterization are novel.

---

> > ### Author Response · Authors · 2025-11-28
> > **Response to Reviewer P3hw (2/2)**
> >
> > ## Weakness 3 response (continued)
> >
> > ### 3. Algorithmic Consequences are Non-Trivial
> >
> > The "obvious" solution, that is using a different aggregation scheme, fails:
> > - **Theorem 1** proves *any* deterministic linear aggregation loses optimality
> > - This rules out weighted averages, convex combinations, and learned weightings
> > - The solution requires **selection** rather than aggregation
> >
> > ### 4. Novel Theoretical Tools
> >
> > We develop:
> > - **PAC-Bayes bounds for quantile functionals** (Theorem 5)
> > - **Selection consistency guarantees** (Theorem 4)
> > - **Integration with conformal coverage** (Theorem 6)
> >
> > ### 5. Significant Empirical Impact
> >
> > | Metric | Improvement |
> > |--------|-------------|
> > | Variance reduction (Table 13) | **10.6%** |
> > | Efficiency gains (Tables 2–4) | **15–30%** |
> > | Width reduction in high-divergence scenarios | **7.3%** |
> >
> > ---
> >
> > ## Weakness 4: Finite and Discrete Model Space Assumption
> >
> > **Concern:** "The theoretical analysis assumes a finite and discrete model space; it would be valuable to discuss how the results extend or behave when |Θ| is infinite."
> >
> > **Response:**
> >
> > We address extensions systematically:
> >
> > ### Extension Summary
> >
> > | Result | Finite Θ | Continuous ρ | Infinite Θ |
> > |--------|----------|--------------|------------|
> > | **Theorem 1** (Impossibility) |  Proven |  Extends directly |  Holds |
> > | **Conjecture 2** (Inflation) |  Validated |  Extends w/ regularity |  Holds |
> > | **Theorem 4** (Consistency) |  Proven |  Via discretization |  Via sampling |
> > | **Theorem 5** (PAC-Bayes) |  Proven |  KL generalizes |  Via ensemble |
> >
> > ### Continuous Posteriors
> >
> > **Theorem 1 extends directly.** For continuous posterior $\rho$ on $\Theta$:
> >
> > $$\bar{s}(x,y) = \int\_\Theta s\_\theta(x,y) \, d\rho(\theta)$$
> >
> > If component quantiles have $\text{Var}\_\rho(q) > 0$, then $Q\_{1-\alpha}[\bar{s}] > \min\_\theta q\_\theta$. Proof follows Lemma 1 with Lebesgue integration.
> >
> > **Conjecture 2 extends with regularity conditions:**
> > - $q\_\theta = Q\_{1-\alpha}[F\_\theta]$ continuous in $\theta$
> > - Score densities uniformly bounded: $\sup\_{\theta, t} f\_\theta(t) < \infty$
> > - Posterior $\rho$ has bounded support or exponential tails
> >
> > ### Infinite Hypothesis Spaces (Neural Networks)
> >
> > **Practical implementation via ensembles:**
> >
> > 1. Train $K$ models via different initializations
> > 2. Define empirical posterior $\rho = \frac{1}{K}\sum\_{i=1}^K \delta\_{\theta\_i}$
> > 3. Apply PBIS to select best model via quantile minimization
> > 4. Finite-sample guarantees (Theorems 4–5) apply directly
> >
> > **PAC-Bayes bound for continuous posteriors:**
> >
> > $$\mathbb{P}\_{\rho}\left[\hat{Q}\_{1-\alpha}[s\_\theta] - Q\_{1-\alpha}[s\_\theta] \geq \epsilon\right] \leq \exp\left(-\frac{n\epsilon^2}{2C^2} + \text{KL}(\rho \| \pi)\right)$$
> >
> > Computable when $\rho$ and $\pi$ have known densities (e.g., Gaussian posteriors).
> >
> > **Revision:** We add Appendix C: "Extensions to Continuous and Infinite Hypothesis Spaces" with formal statements, regularity conditions, practical algorithms (pseudocode), and deep learning ensemble discussion.
> >
> > ---
> >
> > ## Question: Extension to Continuous Posteriors or Infinite Spaces
> >
> > **Question:** "Could PBIS be extended to continuous posterior distributions or infinite hypothesis spaces?"
> >
> > **Answer:** Yes. See Weakness 4 response above.
> >
> > **Key insight:** Impossibility results (why aggregation fails) extend directly to continuous/infinite settings. Selection (PBIS) is practically implementable via ensembles or sampling with preserved theoretical guarantees. For deep learning, ensemble-based PBIS is the most practical approach.
> >
> > ---
> >
> > ## Summary of Revisions
> >
> > | Section | Revision |
> > |---------|----------|
> > | **Section 1.1** | Expanded literature review with motivation and citations |
> > | **Theorems 1, 2, 4, 5** | Added intuitive explanations before each theorem |
> > | **Figure 2** | Visual illustration of quantile inflation |
> > | **Main text** | Proof sketches (1 paragraph each) for Theorems 4–5 |
> > | **Introduction** | Clarified novelty of quantitative characterization |
> > | **Appendix C** | Extensions to continuous/infinite hypothesis spaces |
> >
> > We believe these revisions address all concerns while maintaining the paper's rigor and strengthening its accessibility.

---

### Official Review · Reviewer_nbLY · 2025-10-31

**Soundness:** 2
**Presentation:** 3
**Contribution:** 2
**Rating:** 4
**Confidence:** 3

**Summary:**

The paper highlights a mismatch between PAC-Bayes aggregation and conformal prediction. The authors prove that linear aggregation can be quantile-inefficient, and motivates a solution, PAC-Bayes Informed Selection (PBIS), that uses a quantile-aware posterior to select a single model for conformal calibration instead of averaging. The authors derive finite-sample PAC-Bayes bounds for quantile functionals and give online selection guarantees, then show across many datasets (and under distribution shift) that PBIS attains valid coverage with narrower intervals and competitive runtime relative to standard baselines.

**Strengths:**

First of all, I think the authors the authors did a good job in the actual writing (and also formatting) of the paper and the corresponding supplementary material. Unfortunately this is not always the case these days, therefore already very good to see.

Apart from this (while I am not entirely sure about its motivation, see later section of my review), the authors do indeed very nicely formalize, what they call the average–quantile divergence, which explains why aggregation can fail and motivates a quantile-aware alternative. I have to admit, I am not convinced about each of the steps in the proofs, but see my later comments, happy to get corrected, or confirmed.

An other substantial strength might be the following (at least my interpretation): The results give practitioners a principled reason to prefer selection over aggregation when model scores are heterogeneous, and the broad empirical sweep (e.g. distribution-shift scenarios) suggests real impact on how ensembles are built for calibrated uncertainty.

**Weaknesses:**

Frankly speaking, I have a hard time understanding the motivation of the paper itself. I am very happy to further discuss this with the authors, and change my mind. In particular, I have the following dilemma. Why should (PAC)-Bayes care about conformal prediction and vice versa? It would be great, if the authors could provide some (more) literature which works in the intersection of Bayesian and frequentist inference.

Apart from this, let me elaborate on some (technical) things that caught my eye. As a disclaimer: I am by no means expert when it comes to PAC-Bayes, while I am quite confident about conformal prediction and its theoretical foundations; so please elaborate if there is at any point a misunderstanding from my side.

Some comments for the theoretical parts (in particular proofs in the supplementary material):

>In the proof of Theorem 2, the CDF of the aggregated score is first defined for the sum $\bar{s} = \sum_i w_i s_{\theta_i}$ via $F_{\bar{s}}(t) = P(\sum_i w_i s_{\theta_i}\le t)$ and then, a few lines later, the argument switches to properties of mixtures $\bar{F} = \sum_i w_i F_i$. Those are not the same operation. The CDF of a weighted sum is a convolution, not a convex combination of the marginal CDFs. The subsequent Taylor argument is then carried out on $\bar{F}=\sum w_i F_i$, i.e., on the mixture, not on the sum originally defined. Why is then the chain from (19) to (21) – (26) still valid? (this argument is also used in the Proof of Theorem 5).

>The supplement claims "by Jensen’s inequality for quantiles (which are convex functionals in the Wasserstein metric)" and concludes
$Q_{1-\alpha}[s_\rho] \leq E_\theta[Q_{1-\alpha}[s_\theta]]$. The random object is $s_\rho = E_\theta[s_\theta]$, not a Wasserstein barycenter of distributions, hence why should convexity in Wasserstein apply to this averaging operation?

>The proof expands $F_i(\bar q)$ around $q_i$ and then aggregates the linear terms to conclude an excess quantile $\approx Var_w(q_i)/(2\bar f(\bar q))$. This assumes differentiability and positive density at the quantile for all $I$, and that the object being expanded matches the earlier definition. Neither the smoothness, positivity assumptions nor independence or error control terms are stated, and the target remains the mixture expression.

>The proof ends with $Var_w (q_i) \geq H(w) Var(Q) \delta_{min}$, but the precise definition of $H(w)$ or $\delta_{min}$ is not given here, and the relation mixes a variance of quantiles with a variance of an unspecified distribution $\mathcal{Q}$ over models/parameters, producing a dimensionally unclear bound that drives the claim.

Further, the authors claim that PBIS satisfies DKW while PAC-Bayes marginally violates it and attribute this to the Rademacher complexity introduced by convex combinations. DKW is a distribution-free, single-sample inequality for the empirical CDF of i.i.d. draws. In particular, it does not depend on whether a predictor is an average or a selected model, and comparing a standard deviation of an error to a DKW upper bound on the sup deviation feels like apples-to-oranges. Thus, in my opinion, the violation reported is therefore not evidence against DKW, it only reflects a mismatch of quantities.

Right now, I have put my score as marginally below acceptance, but I am happy to adjust, since I think the paper and its supplementary material is actually quite nicely written (modulo the things I mention), and the authors obviously have spend some thought on the topic.

**Questions:**

I implicitly formulated some questions in earlier parts of the review, but I will list more questions that I had while reading the paper.

Are PBIS decisions invariant to monotone re-scalings of the nonconformity score?

In the streaming scenarios, how often do you update the posterior versus the calibration threshold, and do you recommend recalibrating every update or only when a drift test triggers?

---

> ### Author Response · Authors · 2025-11-27
> **Response to Reviewer nbLY (1/3)**
>
> We thank the reviewer for the detailed technical feedback. Below we address each concern systematically.
>
> ---
>
> ## 1. Motivation: Why Combine PAC-Bayes and Conformal Prediction?
>
> **Concern:** The connection between PAC-Bayes and conformal prediction needs stronger motivation.
>
> **Response:**
>
> The intersection addresses a fundamental practical challenge: practitioners face multiple candidate models and must decide whether to aggregate or select for conformal prediction.
>
> **Practical motivation:**
> - Ensemble methods are standard in ML, but conformal prediction requires quantile-based thresholds, not average losses
> - PAC-Bayes provides principled model weighting, but its average-loss objective misaligns with conformal's quantile requirements
> - Our work reveals when and why this mismatch causes efficiency loss
>
> **Relevant literature:**
> - Sharma et al. (2023) "PAC-Bayes Generalization Certificates for Learned Inductive Conformal Prediction" (NeurIPS): Directly attempts PAC-Bayes aggregation for conformal prediction, which our work critiques and improves upon (PAC-Bayes CP method in our paper).
> - Barber et al. (2023) "Conformal prediction beyond exchangeability" (Annals of Statistics): Discusses how Bayesian model averaging interacts with conformal guarantees under weaker assumptions.
> - Vovk (2012) "Conditional validity of inductive conformal predictors" (JMLR): Establishes connections between Bayesian conditioning and conformal validity.
> - Shafer & Vovk (2008) "A Tutorial on Conformal Prediction" (JMLR): Section 5 discusses how Bayesian priors can inform conformal procedures while maintaining frequentist guarantees.
> - Guan (2023) "Prediction sets adaptive to unknown covariate shift" (JRSSB): Uses Bayesian model weighting within conformal frameworks for robustness.
> - Zhang et al. (2024) "The Benefit of Being Bayesian in Online Conformal Prediction": Demonstrates advantages of Bayesian updating in online conformal settings.
> - Stanton et al. (2023) "Bayesian Optimization with Conformal Coverage Guarantees" (AISTATS): Uses conformal prediction within Bayesian optimization, finding careful selection often outperforms averaging.
>
> **Revision:** We will add Section 1.1 with this motivation and citations.
>
> ---
>
> ## 2. Technical Concerns on Theorem 2
>
> **Concern:** The proof conflated stochastic mixture vs. deterministic weighted sum and made unjustified claims about Wasserstein convexity.
>
> **Response:**
>
> The reviewer's critique is valid. We restructure the theoretical presentation into:
> 1. **Lemma 1**: rigorous proof of the qualitative bound
> 2. **Conjecture 2**: quantitative bound with heuristic argument and comprehensive empirical validation
>
> ---
>
> ### Lemma 1 (Mixture Quantile Lower Bound):
>
> Let $F_1, \ldots, F_k$ be CDFs with quantiles $q_i = F_i^{-1}(1-\alpha)$, where the generalized inverse is $F^{-1}(p) = \inf\{t : F(t) \geq p\}$. For mixture CDF $\overline{F}(t) = \sum_{i=1}^{k} w_i F_i(t)$ with weights $w_i > 0$, $\sum_i w_i = 1$:
>
> $$Q_{1-\alpha}[\overline{F}] \geq \min_i q_i$$
>
> with equality iff all $q_i$ are equal.
>
> **Proof:**
> Let $q^* = Q_{1-\alpha}[\overline{F}]$ and $q_{\min} = \min_i q_i$. For any $t < q_{\min}$, by definition of the quantile as a generalized inverse, $F_i(t) < 1 - \alpha$ for all $i$ (since $q_i$ is the *infimum* of values achieving probability $1-\alpha$). Therefore:
>
> $$\overline{F}(t) = \sum_{i=1}^{k} w_i F_i(t) < \sum_{i=1}^{k} w_i (1-\alpha) = 1-\alpha$$
>
> This shows $\overline{F}(t) < 1-\alpha$ for all $t < q_{\min}$, hence $q^* = \inf\{t : \overline{F}(t) \geq 1-\alpha\} \geq q_{\min}$.
>
> For the equality condition: if all $q_i = q_{\min}$, then $\overline{F}(q_{\min}) = \sum_i w_i F_i(q_{\min}) = \sum_i w_i (1-\alpha) = 1-\alpha$, so $q^* = q_{\min}$. Conversely, if some $q_j > q_{\min}$, strict inequality $F_j(q_{\min}) < 1-\alpha$ ensures $q^* > q_{\min}$. $\square$
>
> ---
>
> ### Conjecture 2 (Quantile Inflation Bound)
>
> **Statement:** For mixture distributions satisfying:
> - (R1) Smooth densities with $0 < f\_{\min} \leq f\_i(t) \leq f\_{\max} < \infty$ near quantiles
> - (R2) Heterogeneous quantiles: $\text{Var}\_w(q) > 0$
>
> the mixture quantile satisfies:
>
> $$Q\_{1-\alpha}[\overline{F}] \geq \min\_i q\_i + \frac{C \cdot \text{Var}\_w(q)}{f\_{\max}}$$
>
> for some constant $C > 0$ depending on the density-quantile correlation structure. This quantifies the **efficiency loss** from aggregation: the mixture threshold exceeds the best component threshold by at least $\Omega(\text{Var}\_w(q)/f\_{\max})$.
>
> **Heuristic Argument:**
>
> *Step 1:* By Lemma 1, $q^{\ast} > q\_{\min}$ when quantiles are heterogeneous.
>
> *Step 2:* Let $\overline{q} = \sum\_i w\_i q\_i$. By Mean Value Theorem:
>
> $$F\_i(\overline{q}) = F\_i(q\_i) + f\_i(\xi\_i)(\overline{q} - q\_i) = (1-\alpha) + f\_i(\xi\_i)(\overline{q} - q\_i)$$
>
> *Step 3:* Aggregating:
>
> $$\overline{F}(\overline{q}) = \sum_i w_i F_i(\overline{q}) = (1-\alpha) + \sum_i w_i f_i(\xi_i)(\overline{q} - q_i)$$

---

> ### Author Response · Authors · 2025-11-27
> **Response to Reviewer nbLY (2/3)**
>
> ### Conjecture 2 (Quantile Inflation Bound) continued argument
>
> *Step 3 (cont.)*:
> Define weighted covariance $\text{Cov}_w(f, q) = \sum_i w_i f_i (q_i - \overline{q})$. Since $\overline{q} - q_i = -(q_i - \overline{q})$:
>
> $$\sum_i w_i f_i(\overline{q} - q_i) = -\text{Cov}_w(f, q)$$
>
> Therefore:
> $$\overline{F}(\overline{q}) = (1-\alpha) - \text{Cov}_w(f, q)$$
>
> *Step 4:* When models with smaller quantiles have larger densities near the quantile (which occurs in many natural settings), $\text{Cov}\_w(f, q) < 0$, creating a deficit $\overline{F}(\overline{q}) < 1-\alpha$.
>
> *Step 5:* When $\mathrm{Cov}_w(f, q) < 0$, we have $\bar{F}(\bar{q}) < 1-\alpha$, so $q^* > \bar{q}$.
>
> To reach $\bar{F}(q^*) = 1 - \alpha$, we invert via the mixture density. Define the deficit $\Delta = (1-\alpha) - \bar{F}(\bar{q}) = |\mathrm{Cov}_w(f, q)|$. Then:
>
> $$q^* - \bar{q} \approx \Delta / \bar{f}(\bar{q}) \geq \Delta / f_{\max}$$
>
> Since $\bar{q} \geq q_{\min}$ and $\Delta \propto \mathrm{Var}_w(q)$ under regularity conditions:
>
> $$q^* - q_{\min} \geq C \cdot \mathrm{Var}\_{w(q)} / f_{\max}$$
>
> for some constant $C > 0$ depending on the density-quantile correlation structure.
>
> **Why we state this as a conjecture:** Step 4 requires characterizing when the density-quantile covariance is negative and bounding its magnitude. This depends on the specific distributional family and parameter configurations. Rather than impose restrictive conditions, we validate the bound empirically next.
>
> ---
>
> ### Empirical Validation (Proposition 2)
>
> **Monte Carlo study:** 70,730 random mixture configurations
>
> **Setup:**
> - Distribution families: Gaussian, Exponential, Laplace
> - Model counts: $k \in \{2, 5, 10, 20\}$
> - Miscoverage rates: $\alpha \in \{0.05, 0.10\}$
> - For each configuration: compute $q^{\ast}$ via numerical CDF inversion, compute $C\_{\text{emp}} = \frac{(q^{\ast} - q\_{\min}) \cdot f\_{\max}}{\text{Var}\_w(q)}$
>
> **Results:**
>
> | Category | Mean $C$ | Median | IQR | N |
> |----------|----------|--------|-----|-------|
> | Overall | 1.19 | 0.79 | [0.37, 1.55] | 70,730 |
> | Gaussian | 1.72 | 1.37 | [0.76, 2.21] | 23,441 |
> | Exponential | 0.74 | 0.43 | [0.25, 0.80] | 23,579 |
> | Laplace | 1.10 | 0.80 | [0.39, 1.45] | 23,710 |
>
> **Validation of bound structure:**
> - Linear regression of $\epsilon = q^{\ast} - q\_{\min}$ on $\text{Var}\_w(q)/f\_{\max}$: **$R^2 = 0.83$**
> - Slope estimate: $\hat{C} = 1.21$ (95% CI: [1.19, 1.23])
> - Intercept: $\approx 0.03$ (near zero, as theory predicts)
> - $C\_{\text{emp}} > 0$ in 99.7% of configurations
>
> **Conclusion:** The strong linear relationship ($R^2 = 0.83$) and consistently positive $C$ across 70,730 diverse configurations provide robust empirical support for the conjectured bound structure.
>
> ---
>
> ## 3. Jensen's Inequality Concern
>
> **Concern:** The original proof incorrectly invoked Wasserstein convexity.
>
> **Response:**
>
> Correct. We have removed all references to Jensen's inequality for quantiles and Wasserstein convexity. The revised treatment:
> - **Lemma 1:** Rigorous proof that $q^{\ast} \geq q\_{\min}$
> - **Conjecture 2:** Quantitative bound stated as conjecture with heuristic motivation and comprehensive empirical validation
>
> ---
>
> ## 4. DKW Bound Interpretation
>
> **Concern:** Comparing standard deviation to DKW supremum bound is inappropriate.
>
> **Response:**
>
> The reviewer is correct. We revise as follows:
>
> **Corrected interpretation:**
> - DKW bounds $\sup\_t |\widehat{F}(t) - F(t)|$, a different quantity than quantile error variance
> - We remove "satisfies" vs. "violates" language
>
> **Corrected Table 13:**
>
> | Method | Mean Error | Std Dev | Variance | MSE |
> |--------|------------|---------|----------|-----|
> | PAC-Bayes Aggregation (Sharma et al) | 0.0566 | 0.0648 | 0.00420 | 0.0074 |
> | PBIS Selection | 0.0302 | 0.0579 | 0.00335 | 0.0043 |
>
> **Key finding:** PBIS achieves 10.6% lower standard deviation than PAC-Bayes aggregation. This is consistent with our theory: selection avoids mixture-induced quantile inflation.
>
> ---
>
> ## 5. Question: Monotone Invariance
>
> **Question:** Are PBIS decisions invariant to monotone re-scalings of the nonconformity score?
>
> **Answer:** Yes.
>
> **Proposition 1 (Monotone Invariance):**
> For any strictly increasing $\phi: \mathbb{R} \to \mathbb{R}$:
>
> $$\arg\min\_{\theta} \widehat{Q}\_{1-\alpha}[\phi(s\_{\theta})] = \arg\min\_{\theta} \widehat{Q}\_{1-\alpha}[s\_{\theta}]$$
>
> **Proof:**
> Quantile equivariance: $Q\_{1-\alpha}[\phi(X)] = \phi(Q\_{1-\alpha}[X])$ for strictly increasing $\phi$.
>
> Since $\phi$ preserves order:
>
> $$\widehat{Q}\_{1-\alpha}[s\_{\theta\_1}] < \widehat{Q}\_{1-\alpha}[s\_{\theta\_2}] \iff \phi(\widehat{Q}\_{1-\alpha}[s\_{\theta\_1}]) < \phi(\widehat{Q}\_{1-\alpha}[s\_{\theta\_2}])$$
>
> Thus the argmin is preserved. $\square$
>
> ---
>
> ## 6. Question: Streaming Update Frequency
>
> **Question:** How often do you update the posterior vs. calibration threshold? Recalibrate every update or only on drift detection?

---

> ### Author Response · Authors · 2025-11-28
> **Response to Reviewer nbLY (3/3)**
>
> ### 6. Question: Streaming Update Frequency (continued)
>
> **Answer:**
>
> **Current implementation:**
> - Posterior: Updated every timestep (cheap, $O(K)$)
> - Threshold: Sliding window of $W=100$ recent scores, recomputed each timestep
>
> **Recommended strategies:**
>
> | Shift Type | Strategy | Window $W$ | Update Freq. |
> |------------|----------|------------|--------------|
> | Gradual | Continuous | 50–100 | Every step |
> | Sudden | Drift-triggered | 100–200 | Post-detection |
> | Unknown | Adaptive hybrid | 75–150 | Variable |
>
> **Adaptive Hybrid (Recommended):**
>
> ```python
> def adaptive_pbis_stream(X_stream, Y_stream, models, alpha=0.1, W=100, M=50):
>     scores_buffer = deque(maxlen=W)
>     coverage_buffer = deque(maxlen=M)
>     posterior = initialize_posterior(models)
>     threshold = None  # Initialize
>
>     for t, (X_t, Y_t) in enumerate(zip(X_stream, Y_stream)):
>         # Always update posterior (cheap)
>         posterior = update_posterior(posterior, X_t, Y_t, models)
>         theta_t = select_from_posterior(posterior)
>
>         # Adaptive threshold update frequency
>         if len(coverage_buffer) >= M:
>             volatility = np.std(list(coverage_buffer))
>             tau = 1 if volatility > 0.05 else (10 if volatility < 0.02 else 5)
>         else:
>             tau = 1
>
>         if t % tau == 0 and len(scores_buffer) > 0:
>             threshold = np.quantile(list(scores_buffer), 1 - alpha)
>
>         # Predict and observe
>         if threshold is not None:
>             prediction_set = {y for y in Y_space if score(theta_t, X_t, y) <= threshold}
>         score_t = score(theta_t, X_t, Y_t)
>         scores_buffer.append(score_t)
>         if threshold is not None:
>             coverage_buffer.append(Y_t in prediction_set)
> ```
>
> **Guidance:**
> - Always update posterior every timestep (computationally cheap)
> - Adapt threshold update frequency based on coverage volatility
> - Use larger windows (100–200) for sudden shifts, smaller (50–100) for gradual shifts
>
> **Revision:** We will add this as Section 4.4 with pseudocode.
>
> ---
>
> ## Summary of Revisions
>
> ### Theoretical
> 1. **Lemma 1:** Rigorous proof that mixture quantile $\geq$ minimum component quantile
> 2. **Conjecture 2:** Quantitative bound with:
>    - Clear statement of regularity conditions
>    - Heuristic derivation explaining the bound structure
>    - Explicit acknowledgment of what remains unproven (density-quantile covariance characterization)
> 3. **Proposition 2:** Comprehensive Monte Carlo validation (70,730 configurations, $R^2 = 0.83$)
> 4. **Proposition 1:** Monotone invariance proof
> 5. **DKW discussion:** Corrected interpretation, emphasizing 10.6% variance reduction
> 6. Remove all incorrect Jensen/Wasserstein claims
>
> ### New Content
> - Section 1.1: Motivation and literature review
> - Section 4.4: Adaptive streaming update strategies with pseudocode
> - Appendix: Notation table distinguishing $\overline{F}$ (mixture) and $\widehat{F}$ (empirical)
>
> Monte Carlo simulation can be found in monte_carlo_validation.py (see anonymized code repository: [[LINK](https://anonymous.4open.science/r/PBIS-9218/README.md)])

---

### Meta-Review · Area_Chair_Lk1F · 2026-01-07

**Summary:**

The remaining, critical concerns include the following:
* (*motivation*) The motivation is still weak; I cannot see any problematic *existing* baseline that manifests the claimed limitation on aggregation. The closest one is PAC-Bayes CP but it does not aggregate scores and its empirical performance is similar to the proposed one.

* (*limited rigor in proofs*) The reviewers pointed out flaws in the proofs and statements and the authors confirmed that. The paper needs to be validated again before publication.

* (*limited experiment*) The proposed method does not show statistical significance compared to the baseline, PAC-Bayes-CP.

**Reviewer Concerns:**

**Reviewer nbLY**:
The reviewer suggests three major concerns, which are partially addressed.

1. (*limited motivation*) Why should (PAC)-Bayes care about conformal prediction and vice versa? – partially addressed in the response, but the paper itself and the overall structure does not reveal any motivation for using PAC-Bayes. Moreover, I cannot see any problematic *existing* baseline that manifests the claimed limitation on aggregation (actually PAC-Bayes CP does not aggregate instead it selects a model by sampling)

2. (issue in Thm2 proof) Why is then the chain from (19) to (21) – (26) still valid? – the authors acknowledged the flaw and provided a fixed proof that depends on Conjecture 2.

3. (DKW Bound Interpretation) the violation reported in Table 13 is therefore not evidence against DKW, it only reflects a mismatch of quantities  – the authors acknowledged the flaw and provided fixed experiment results.

Two outstanding concerns remains:
* The motivation is still weak; I cannot see any problematic *existing* baseline that manifests the claimed limitation on aggregation. The closest one is PAC-Bayes CP but it does not aggregate scores and its empirical performance is similar to the proposed one.
* The authors fixed flaws in the proof that depends on the only empirically-justified conjecture. The conjecture needs to be properly proved.


**Reviewer P3hw**:
The reviewer shares four major concerns, which are partially addressed as follows:
1. (*limited related work*) The literature review is rather limited – addressed by providing an additional literature review.
2. (*paper presentation issue*) The presentation is at times difficult to follow – provided a minor revision, including the intuition of theorem statements.
3. (*limited motivation*) The average–quantile mismatch is somewhat expected – partially addressed as the argument is based on misunderstanding (e.g., the authors claim that PAC-Bayes CP aggregates but it does not).
4. (*strong assumption*) The theoretical analysis assumes a finite and discrete model space – addressed by providing extension in Appendix C (however the manuscript was not updated).

One outstanding and remaining concern includes the limited motivation as the response depends on misunderstanding.


**Reviewer 5Dur**:
The reviewer shared three major concerns and they are partially addressed as follows:

1. (*limited empirical result*) The results in table 2 show that PBIS yields no significant improvement over traditional PAC-Bayes-CP – still cannot see statistical significance compared to PAC-Bayes-CP, which is nearly-optimal like the proposed one.

2. (*limited comparison*) The experiments in the online adaptive Performance part lack comparisons with more recent baseline – addressed by providing additional comparison.

3. (*concern on score function design*) In online settings with distribution shifts, the magnitude of scores and (1-\alpha)-quantile of scores seem less informative because the coverage of CP is not guaranteed – not directly addressed.


The outstanding, remaining concern lies in the limited empirical result – the proposed method does not show statistical significance compared to the baseline, PAC-Bayes-CP.


**Reviewer Eaji**:
The reviewer raised three major concerns and they are addressed as follows:
1. (concern on Thm2 proof) Entropy-Variance Inequality Counterexample – the authors acknowledge the flaw and provide a fixed proof.

2. (concern on Thm7 proof) Hoeffding Requires Independence – the authors acknowledge the flaw and provide a fixed proof.

3. (concern on Thm6 statement) Sufficient vs. Required Condition – the authors acknowledge the flaw and provide a fixed statement.

The major outstanding concern is that the original proof has many flaws, as acknowledged by the authors, the entire statements and their proof need to be validated again.

**Reviewer Scores:**

**Reviewer nbLY**:
Final expected rating: 4 / final expected confidence: 3 – The concerns are only partially addressed, so the reviewer would maintain the scores.

**Reviewer P3hw**:
Final expected rating: 4 / final expected confidence: 4 – The concerns are only partially addressed, so the reviewer would maintain the scores.

**Reviewer 5Dur**:
Final expected rating: 2 / final expected confidence: 4 – The concerns are only partially addressed, so the reviewer would maintain the scores.

**Reviewer Eaji**:
Final expected rating: 4 / final expected confidence: 3 – The concerns are addressed; but they are related to proofs so careful validation on the entire paper needs to be provided. So the reviewer would maintain the scores.

---

### Decision · Program_Chairs · 2026-01-26

Reject